# Rapid neo-sex chromosome evolution and incipient speciation in a major forest pest

Ryan R. Bracewell[1,5], Barbara J. Bentz[2], Brian T. Sullivan[3] & Jeffrey M. Good[4]

Genome evolution is predicted to be rapid following the establishment of new (neo) sex chromosomes, but it is not known if neo-sex chromosome evolution plays an important role in speciation. Here we combine extensive crossing experiments with population and functional genomic data to examine neo-XY chromosome evolution and incipient speciation in the mountain pine beetle. We find a broad continuum of intrinsic incompatibilities in hybrid males that increase in strength with geographic distance between reproductively isolated populations. This striking progression of reproductive isolation is coupled with extensive gene specialization, natural selection, and elevated genetic differentiation on both sex chromosomes. Closely related populations isolated by hybrid male sterility also show fixation of alternative neo-Y haplotypes that differ in structure and male-specific gene content. Our results suggest that neo-sex chromosome evolution can drive rapid functional divergence between closely related populations irrespective of ecological drivers of divergence.

[1] Department of Ecosystem and Conservation Sciences, The University of Montana, 32 Campus Drive, Missoula, MT 59812, USA. [2] USDA Forest Service Rocky Mountain Research Station, 860 North 1200 East, Logan, UT 84321, USA. [3] USDA Forest Service Southern Research Station, 2500 Shreveport Highway, Pineville, LA 71360, USA. [4] Division of Biological Sciences, The University of Montana, 32 Campus Drive, Missoula, MT 59812, USA. [5] Present address: Department of Integrative Biology, University of California, Berkeley, CA 94720, USA. Correspondence and requests for materials should be addressed to R.R.B. (email: ryan.bracewell@berkeley.edu) or to J.M.G. (email: jeffrey.good@umontana.edu)

Understanding how reproductive isolation evolves between populations is essential to understanding the origin of biodiversity. Nearly a century of research has established that hybrid incompatibilities usually manifest in the hetero-gametic (male) sex first (i.e., Haldane's Rule[1]) and that the early stages of hybrid male sterility (HMS) and inviability (HMI) are almost always asymmetric in reciprocal genetic crosses[2, 3]. These patterns reflect the fundamental role that heteromorphic sex chromosomes play in the evolution of deleterious epistatic interactions underlying intrinsic reproductive isolation[4–6], but the importance of these processes to the early stages of speciation has been debated[7]. For example, while sex-linked hybrid sterility generally evolves much more rapidly than inviability, both are thought to accumulate more slowly than other forms of ecological or behavioural reproductive isolation[2, 8]. However, most speciation research has focused on animal systems with well-established and highly heteromorphic sex chromosomes[6, 9], resulting in a limited view of the contribution of sex chromosome evolution to the origin of species.

It is now apparent that transitions to new (neo) sex chromosomes are common over evolutionary timescales[10–12], with some groups showing high levels of turnover in chromosomal systems of sex determination[10, 11, 13]. Neo-sex chromosomes can originate through diverse mechanisms, such as the de novo evolution of a sex determining factor on otherwise homomorphic autosomal chromosomes (e.g., wild strawberry[14]) or through the fusion of an ancestral sex chromosome with an autosome (e.g., *Drosophila miranda*[15]). Given the surprisingly fluid nature of sex chromosome systems in some groups, there are at least three general stages of sex chromosome evolution—each with distinct evolutionary dynamics—that might contribute to the process of speciation (Fig. 1). First, the establishment of a new sex chromosome system within a population could lead to reproductive isolation between species with different sex chromosome systems. Second, once established, the functional conversion of autosomes to sex chromosomes results in dynamic changes in chromosome structure, gene content, and expression underlying the evolution of sex-specific functions[15–17]. The extensive genic specialization and structural degeneration accompanying such chromosomal transformations provide some of the most extreme examples of long-term genome evolution[18]. In XY systems, the rate of sex chromosome differentiation and neo-Y degeneration is dependent on the suppression of recombination between the neo-XY pair[15, 16]. In principle, sex chromosome transitions could also result in rapid functional divergence and the accumulation of genetic incompatibilities between populations, but the tempo of these genomic changes and their contribution to broader patterns of species diversification remains largely unknown. For example, genes involved in reproductive isolation have been linked to recently established neo-sex chromosomes in threespine stickle-backs[19] and butterflies[20], but it is unclear if or how these hybrid incompatibilities relate to neo-sex chromosome evolutionary dynamics per se. Finally, it is well established that older, highly heteromorphic sex chromosomes often play a disproportionately large role in speciation through the exposure of recessive genetic incompatibilities in hybrid males (i.e., dominance theory)[4] and various evolutionary dynamics (e.g., faster-X, meiotic drive) that can drive rapid sex-linked divergence[5, 6].

The mountain pine beetle (*Dendroctonus ponderosae*) is the most important forest pest in western North America, and is the primary contributor to recent tree mortality on 6.6 million hec-tares in the western US, exceeding tree mortality caused by wildfire[21]. Previous studies have found that populations at the southern reaches of the beetle's range (southern California and Arizona) are the most genetically divergent and genetic variation follows an isolation-by-distance pattern around the Great Basin

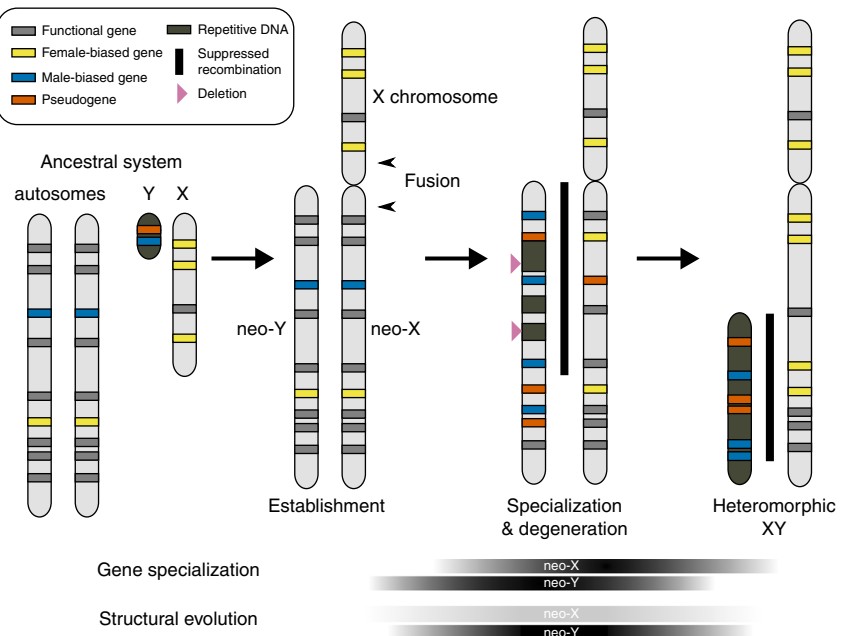

**Fig. 1** Neo-sex chromosome evolution. The predicted stages of heteromorphic differentiation are shown left to right for a neo-XY pair formed from a fusion between an X chromosome and an autosome. The relative timing and intensity of sex-specific gene specialization and structural evolution are depicted for the neo-X and neo-Y separately (shaded boxes, bottom). Following establishment, the neo-sex chromosomes are expected to undergo sex-biased gene specialization and at least some structural evolution dependent on the cessation of recombination. Both processes should occur more rapidly and be more extensive on the male-specific neo-Y chromosome. As seen in highly heteromorphic sex chromosomes, the evolution of heteromorphic neo-sex chromosomes is also predicted to further contribute to reproductive isolation through the exposure of recessive genetic incompatibilities and more rapid sex-linked divergence in the heterogametic sex. The fate of the highly degenerate ancestral Y chromosome is not shown and is often unclear in X-autosome fusions. Figure adapted from Bachtrog[50]

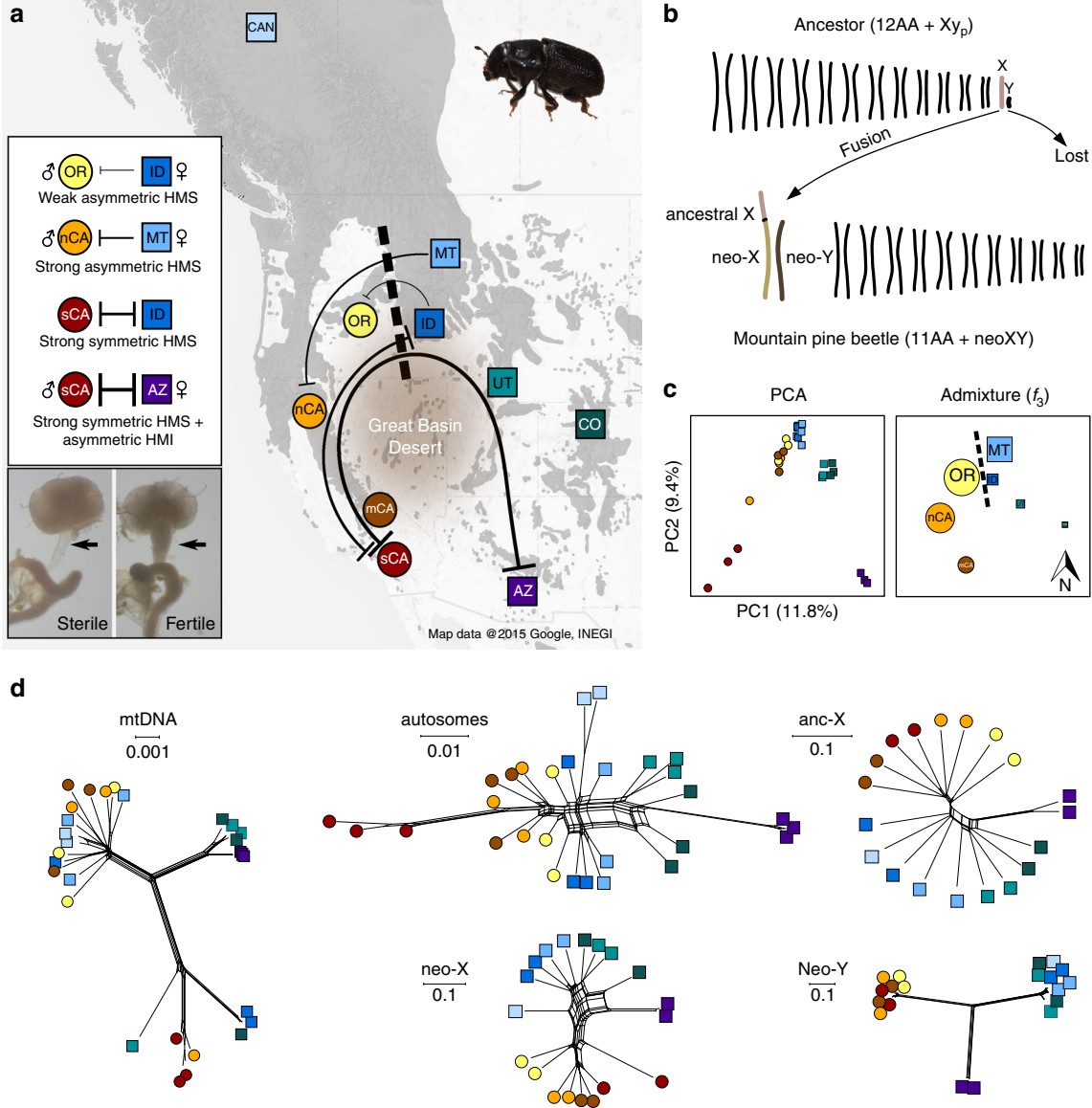

**Fig. 2** Reproductive isolation and genomic differentiation. **a** The mountain pine beetle occurs in close association with host trees in western North America (dark grey) and gene flow is thought to be restricted across the Great Basin Desert. We combined extensive genetic crossing data with beetle genomes sequenced from ten localities (circles and squares; all relevant crossing shown in Supplementary Fig. 1). Reproductive isolation was observed between localities connected with lines, increasing in severity (top left inset) with distance from a central boundary (dashed line; See also Supplementary Fig. 2 and Supplementary Table 1). Sex symbols denote the direction of the cross when any asymmetric isolation (HMS or HMI) was observed (top left inset). Sterile males show reduced quantities of sperm in the seminal vesicle (bottom left inset; see also Supplementary Fig. 3). **b** Model of neo-sex chromosome evolution in the mountain pine beetle (and *D. jeffreyi*) from an ancestral X-autosome fusion. The ancestral $Xy_p$ notation describes systems where the Y chromosome is small and highly degenerate and the X forms a 'parachute' like structure while pairing at a distance from the Y during meiosis[12]. **c** Results from principal component (left) and admixture analyses ($f_3$ statistics, right) for autosomal SNPs across all population comparisons. Significantly admixed populations are positioned by geography and symbol size scaled relative to the number of significant three-population tests (Supplementary Table 3) supporting admixture of the focal population (z-score < −10). **d** SplitsTree networks for mtDNA and four nuclear genomic partitions (autosomes = 60 K SNPs, anc-X = 87 K, neo-X = 75 K, neo-Y = 544). For additional analyses of population structure see Supplementary Fig. 7 and Supplementary Table 2

Desert[22] (although small isolated populations do occur across the Great Basin[23]). Despite very low levels of genetic differentiation between populations[22], hybrid male sterility (HMS) has been detected between some mountain pine beetle populations consistent with cryptic speciation[24]. In addition, the mountain pine beetle shares a neo-XY chromosome system with its closest relative, the Jeffrey pine beetle (*Dendroctonus jeffreyi*), descending from a fusion between the ancestral X chromosome (anc-X) and the largest autosome in the common ancestor of these species[25].

There is no known karyotypic variation within either species[25] and the degree of genetic, structural, or functional divergence between these nascent sex chromosomes has not been evaluated. However, several *Dendroctonus* species and many other beetles have distance-pairing heteromorphic sex chromosomes that do not synapse and are fully non-recombinant[12]. In these systems, the Y chromosome appears highly degenerate and is a small fraction of the size of the X. If distance-pairing and the associated loss of recombination carries over following an autosomal X

chromosome fusion event, then neo-sex chromosomes may be immediately predisposed to the rapid evolution of heteromorphy in this system.

Here we use extensive genetic crossing experiments coupled with population and functional genomics to examine neo-sex chromosome evolution and incipient speciation in the mountain pine beetle system. We detected extraordinary variation in the strength and pattern of reproductive isolation between very closely related populations. The distinct genetic architecture of $F_1$ reproductive isolation in these crosses combined with genomic patterns of population differentiation and admixture on and off the sex chromosomes supports a central role for sex-linked reproductive isolation in mountain pine beetle populations. In parallel, we also document extensive variation in neo-Y structure and male-specific gene content between populations isolated by hybrid male sterility. Overall, our results reveal that extensive genic specialization and structural degeneration on the neo-sex chromosomes can be sufficiently rapid to drive intrinsic functional differentiation between closely related populations, which may in turn play an important role in incipient speciation.

## Results

**A geographic progression of intrinsic reproductive isolation.** We have completed an extensive genetic crossing experiment to characterize patterns of reproductive isolation between populations sampled from across the US range of the mountain pine beetle (Fig. 2a). Combining extensive new (Supplementary Table 1) and published data[24] (9 populations, 18 cross-types, 1109 crosses, Supplementary Fig. 1), we uncovered a geographic continuum of intrinsic hybrid incompatibilities. Crosses between some populations in the central part of the range resulted in HMS (Supplementary Fig. 2) due to disrupted spermatogenesis (Supplementary Fig. 3). Although we have not described hybrid male fertility in all possible population pairs, the severity and $F_1$ architecture of HMS consistently changed with geographic distance of hybridizing population pairs found on either side of a central East-West boundary of reproductive isolation. Crosses between proximate populations on either side of this boundary resulted in hybrid males with reduced fertility (Supplementary Fig. 2) in one direction of the cross (i.e., weak asymmetric HMS; female ID×male OR), while reciprocal crosses between more distant localities resulted in severe or complete HMS (i.e., strong symmetric HMS; Fig. 2a).

We also detected asymmetric delayed hybrid male development in crosses between beetles from Arizona (AZ) and Southern California (female AZ×male sCA; Supplementary Fig. 4), which is the most geographically distant cross type based on the distribution of host trees (Fig. 2a). This transgressive hybrid phenotype likely reflects intrinsic developmental incompatibilities (i.e., HMI), but the fitness consequences depend on population ecology (i.e., extrinsic). Mountain pine beetles are univoltine (one generation per year) and the timing of their development is likely under strong selection in nature[26]. Following emergence, beetles colonize heavily defended host pine trees by staging highly coordinated attacks and development time and emergence synchrony are crucial for beetle success. Thus, delayed development time would be highly maladaptive, thereby providing an ecologically relevant measure of hybrid viability. Information on hybrid development time was not available for all crosses, but available data (Supplementary Fig. 1) suggest that abnormal hybrid male development may be restricted to crosses between the most geographically distant and genetically divergent populations. These populations (AZ and sCA) do not come into contact, thus delayed development may not currently contribute to reproductive isolation in nature but it does reveal another

important component of evolutionary divergence leading to intrinsic incompatibilities between beetle populations.

These cryptic patterns of reproductive isolation (Fig. 2a) did not coincide with described phenotypic differences between populations[27] or follow any obvious ecological gradients (e.g., host use), as might be expected if hybrid incompatibilities are an indirect consequence of ecological divergence. For example, beetles from populations bordering the HMS boundary (OR and ID) were collected from the same host tree species (*Pinus contorta*), were morphologically indistinguishable, and do not differ in body size or development time[27]. Pheromone profiles were also found to be very similar between reproductively isolated OR and ID populations (Supplementary Fig. 5). Although we did observe some slight clustering of female pheromone blends by population (Supplementary Fig. 5), the partial separation that we observed was due to two compounds (acetophenone and 1-phenylethanol) that do not attract either sex to traps or alter responses to an attractive pheromone trap lure[28]. Cumulatively, there is no evidence of ecological divergence between populations near the HMS boundary and the minor pheromone differences that we detected are unlikely to have a meaningful effect on behaviour or mate recognition. Thus, hybrid male sterility and delayed development are the only known phenotypes isolating populations of mountain pine beetles, and the geographic pattern of these intrinsic incompatibilities suggests a fairly complex and polymorphic genetic basis[9].

**Reproductive isolation and population differentiation.** Our extensive crossing experiments revealed a broad range of intrinsic $F_1$ incompatibilities, establishing that reproductive isolation unequivocally proceeds through asymmetric postzygotic isolation (Fig. 2a). Furthermore, there is no evidence for variable endo-symbiont infections (e.g., *Wolbachia*) that might contribute to asymmetric HMS in this system based on standard genetic and molecular diagnostics[24]. Thus, this distinct genetic architecture requires epistatic interactions involving genetic factors with uni-parental inheritance[3]. In mountain pine beetles this equates to nuclear-cytoplasmic interactions involving mitochondrial DNA (mtDNA) and/or sex chromosome-linked interactions. To test this hypothesis and to evaluate the overall genomic context of these hybrid incompatibilities we first generated a range-wide population genomic data set by combining two published genomes[29] (CAN) with 27 additional whole genomes from nine populations (two males/one female sequenced per population, 18× average coverage per genome; Fig. 2a), and two Jeffrey pine beetle genomes (16× average coverage per genome). The published male and female reference genomes identified part of the ancestral X (anc-X) chromosome based on synteny with flour beetles (*Tribolium*), but did not differentiate the autosomes from the neo-sex chromosomes[29] which are thought to descend from an ancestral X-autosome fusion[25] (Fig. 2b). To overcome these limitations, we assembled individual mtDNA genomes and used sex-specific patterns of whole genome sequencing (CAN) to identify autosomal (70.5% of the male genome build), X-linked (16% anc + neo-X; 664 scaffolds), and neo-Y linked (6.6%, 2272 scaffolds) regions of the genome (Supplementary Fig. 6).

These analyses revealed three important attributes of neo-sex chromosome evolution in this system. First, the sex chromosomes comprise ~22% of the assembled genome. This estimate is broadly consistent with karyotypic data[25] and excludes any residual pseudoautosomal regions, which would be indistinguishable from autosomes with respect to sex-specific coverage. Second, the neo-Y appears smaller, contains fewer genes, and is more fragmented when compared to the neo-X chromosome (Table 1, Supplementary Fig. 6). Third, retained homologous neo-

**Table 1 Genome assembly statistics by genomic partition**

| Partition | N50 (kb) | Largest scaffold (kb) | Number of scaffolds | Mean phred-scaled quality score (male)[a] | Number of predicted genes[b] |
|---|---|---|---|---|---|
| neo-Y | 11 | 67 | 2272 | 35.0 | 781 |
| neo-X | 164 | 688 | 658 | 45.4 | 1227 |
| anc-X | 3493 | 3885 | 6 | 55.3 | 666 |
| autosomes | 1024 | 4163 | 1178 | 54.5 | 9308 |

[a]Mean mapping qualities of CAN male reads mapped to male reference genome[29]
[b]On the basis of the predicted annotations from the published assembly[29]

X and neo-Y genic regions show moderate levels of sequence divergence (119 pairwise X–Y gametologs; mean $Ks = 5.7\%$, median $Ks = 2.2\%$). These estimates of synonymous gene divergence are about two-fold higher than those found in *D. miranda*, an early neo-XY system ($Ks = 1.5\%$; ~1 million years old, ~10 generations per year) where signatures of specialization and degeneration are already apparent[15]. The age of the *Dendroctonus* neo-sex chromosomes are not clear, but the Jeffrey pine beetle and mountain pine beetle share the same neo-XY configuration. Assuming a simple mtDNA molecular clock[30], the neo-sex chromosomes in the mountain pine beetle are at least 2–3 million years old (1 generation per year).

We found very low levels of sequence divergence across the genome punctuated by strikingly different patterns of genetic variation within and between populations across different nuclear partitions (autosomes, anc-X, neo-X, and neo-Y) and individual mtDNA genomes (Fig. 2c, d, Supplementary Fig. 7). On the basis of 5.3 million autosomal single nucleotide polymorphisms (SNPs), we found no evidence for a genetic split coincident with the HMS boundary (OR vs. ID $F_{st} = 0.01$). Instead, genetic variation largely grouped populations based on geographic proximity (Fig. 2c) with low overall genetic divergence ($D_{xy}$ average = 0.72%, min = 0.62%, max = 0.83%) and low between-population genetic differentiation ($F_{st}$ average = 0.17; min = 0.00, max = 0.45; Supplementary Table 2). Autosomal phylogenetic networks showed extensive reticulation (Fig. 2d) and estimates of population structure indicate a broad transition in population assignment across the HMS boundary (Supplementary Fig. 7). The mtDNA genomes also showed low sequence divergence between populations with no phylogeographic structure associated with populations isolated by HMS (Fig. 2d). These autosomal and mtDNA results are consistent with previous studies in the mountain pine beetle which found no evidence for a genetic split near the HMS boundary and described isolation-by-distance around the Great Basin Desert with populations at the southern extreme of the distribution being the most genetically divergent[22, 31].

In contrast, we detected markedly increased genetic differentiation on the anc-X and neo-X and extreme differentiation of the neo-Y between reproductively isolated populations (Fig. 2d, Supplementary Fig. 7). On the neo-Y, we found exceptionally low population-level variation in 'East' (CO, UT, ID, and MT), 'West' (sCA, mCA, nCA, and OR) and 'Southeast' (AZ) groups on the neo-Y, such that the vast majority of SNPs corresponded to fixed differences between these three neo-Y haplotypes. While comparisons between partitions are complicated by inherent differences in sequence complexities and assembly qualities, general population genomic patterns presented here and below were consistent across a range of SNP filtering regimes (Methods section).

Combining these broad population genomic patterns with our extensive crossing data provides two key insights into speciation. First, hybrid incompatibilities are accumulating extremely rapidly in this system. Intrinsic reproductive isolation tends to follow predictable patterns that are recapitulated in our crosses: incompatibilities manifest in the heterogametic sex first[1], sterility evolves more rapidly than inviability[2], and asymmetry precedes reciprocal isolation[3]. These evolutionary transitions have mostly been established through comparative analyses of crosses between different species pairs spanning a broad range of genetic divergences[2]. The existence of population-level variation in hybrid incompatibilities is now well established[32]; however, the striking progression in the strength and pattern of intrinsic reproductive isolation occurring between populations separated by such low levels of genomic divergence is highly unusual. Second, sex-linked incompatibilities are playing a central role in the evolution of reproductive isolation. Based on overall patterns of population genetic differentiation and HMS in reciprocal crosses, involvement of mtDNA can be unambiguously dismissed while the X and neo-Y chromosomes show moderate to complete differentiation between populations that are also isolated by hybrid male sterility (Fig. 2a). In particular, we have detected some degree of hybrid male sterility in all crosses between populations with highly divergent West vs. East or Southeast neo-Y haplotypes, and no reproductive isolation between populations bearing the same neo-Y haplotypes (Fig. 2d, Supplementary Fig. 1). Male fertility has not been thoroughly characterized between East and Southeast neo-Y beetles (see below). Given this limitation, we focus subsequent results and discussion on the East vs. West reproductive isolation boundary.

**Population history and natural selection.** The progression of reproductive isolation coupled with low autosomal differentiation around the Great Basin Desert could be produced by two alternative population histories. First, genetic divergence may have accumulated along a series of interbreeding populations in the absence of reduced gene flow (i.e., primary divergence-with-gene-flow). Alternatively, reproductive isolation may have evolved during an initial period of geographic isolation or allopatry followed by secondary contact. The low population differentiation and extensive network reticulation observed at autosomal loci (Fig. 2c, d, Supplementary Fig. 7) could reflect the persistence of shared variation between recently diverged populations (i.e., incomplete lineage sorting independent of secondary gene flow) or secondary contact and introgressive hybridization. Distinguishing between these alternatives is crucial to understanding the evolutionary history and genetic architecture of reproductive isolation in this system.

Interbreeding between two previously isolated populations results in discernible patterns of admixed genetic ancestry[33, 34], allowing us to differentiate these primary vs. secondary models of divergence-with-gene-flow[35]. We first calculated the $f_3$ statistic[34] for all possible three-way population comparisons using 1,269,066 high quality autosomal SNPs. This approach uses allele frequency correlations between populations to provide a formal test of admixture that is robust across a broad range of population histories[33, 34]. We detected strongly negative skews in the $f_3$

**Table 2 Genetic diversity and divergence by genomic partition**

| Partition | Diversity ($\pi$) West | Diversity ($\pi$) East | $D_{xy}$ (West vs East) | $D_{xy}$ (to *D. jeffreyi*) | RND | Diversity relative to autosomes |
|---|---|---|---|---|---|---|
| neo-Y | 0.02% (0.02–0.03) | 0.03% (0.02–0.03) | 0.34% (0.33–0.35) | 2.99% (2.90–3.08) | 0.113 (0.107–0.121) | 2.4% |
| neo-X | 0.24% (0.23–0.24) | 0.18% (0.17–0.18) | 0.37% (0.37–0.38) | 2.63% (2.62–2.62) | 0.142 (0.141–0.145) | 22.5% |
| anc-X | 0.31% (0.31–0.31) | 0.29% (0.29–0.30) | 0.39% (0.38–0.39) | 1.43% (1.42–1.44) | 0.270 (0.263–0.271) | 60.4% |
| autosomes | 0.72% (0.72–0.72) | 0.63% (0.63–0.63) | 0.72% (0.72–0.72) | 1.94% (1.93–1.94) | 0.373 (0.373–0.371) | — |
| mtDNA | 0.72% (0.64–0.81) | 0.75% (0.66–0.84) | 0.80% (0.71–0.88) | 6.54% (6.17–6.92) | 0.122 (0.102–0.143) | 32.3% |

Diversity ($\pi$) and divergence ($D_{xy}$) estimates by linkage category for West and East beetles and divergence ($D_{xy}$) to the outgroup, *Dendroctonus jeffreyi*. Relative node depth (RND[39]) is $D_{xy}$ between the East and West populations divided by $D_{xy}$ to the outgroup. Diversity relative to the autosomes is shown after accounting for differences in mutation rates following Wilson Sayres et al.[38]. Bootstrapped 95% confidence intervals shown in parentheses

statistic in the centre of the range, indicative of secondary contact followed by extensive autosomal admixture between populations adjacent to the HMS boundary with signatures of admixture tailing off with geographic distance from the boundary (Fig. 2c, Supplementary Table 3). These strong signatures of autosomal admixture establish a history of secondary contact near the HMS boundary with subsequent gene flow.

The population-frequency based $f_3$ statistics are not appropriate for sex-linked SNPs given our sampling, so we next tested for introgression at both autosomal and X-linked SNPs using the ABBA-BABA framework[36]. Consistent with the $f_3$ results, we again found evidence of autosomal gene flow near the HMS contact zone (Supplementary Table 4). We detected some gene flow occurring on both the neo-X and anc-X (Supplementary Table 4). These results are in stark contrast to what we observed on the neo-Y chromosome where alternative neo-Y haplotypes appear fixed across the HMS contact zone (i.e., $F_{st}$ is ~1 and 97% of sites are BBAA).

The persistence of genomic divergence in the face of gene flow is often used to infer the genetic basis of reproductive isolation[35], especially in systems where reproductively isolating phenotypes and their underlying genetic architectures are unknown. In mountain pine beetles, the distinct asymmetric architecture of HMS (Fig. 2a) coupled with heterogeneous patterns of differentiation and admixture suggests that the neo-Y and the neo-X chromosomes are involved in the evolution of HMS. Both sex chromosomes also stood out in population comparisons for showing highly elevated $F_{st}$, however, it would be premature to interpret these patterns as entirely products of reduced gene flow[37]. Measures of relative divergence, such as $F_{st}$, are sensitive to levels of variation within and between populations and both sex chromosomes showed reduced population-level diversity ($\pi$) and lower absolute sequence divergence ($D_{xy}$) between populations (Table 2). Thus, elevated sex-linked $F_{st}$ between East and West populations could reflect the combined effects of reduced gene flow as well as any process that reduces variation within these populations (e.g., linked natural selection)[37]. Genetic drift is generally predicted to be stronger on the sex chromosomes due to differences in their effective population sizes. Assuming equal sex ratios, genetic diversity on the X and Y should be ~75 and ~25% of autosomal variation. However, we found that the anc-X, neo-X, and neo-Y all showed significantly reduced levels of genetic variation relative to these theoretical predictions (~60, ~23 and 2% of autosomal diversity, respectively) after accounting for variation in mutation rate (Table 2).

Skews in mating sex ratios cannot account for reduced variation on both sex chromosomes, or for significant reductions in X-linked but not mtDNA diversity. Alternatively, recurrent positive and purifying natural selection can reduce chromosome-wide levels of genetic diversity, especially on non-recombinant Y chromosomes[38]. Absolute sequence divergence (e.g., $D_{xy}$) should be robust to the influence of selection within contemporary populations, but is sensitive to chromosomal differences in effective population sizes including the effects of selection within ancestral populations[37]. We observed lower absolute X and Y-linked divergence ($D_{xy}$) between reproductively isolated populations of beetles (Table 2) and these patterns persisted when $D_{xy}$ was normalized by divergence to the Jeffrey pine beetle to account for variation in substitution rates between chromosomes (i.e., relative node depth or RND[39]). Thus, patterns of diversity and divergence both indicate significant long-term reductions in the effective population sizes of the sex chromosomes in mountain pine beetles. Moreover, while more efficacious selection is common on sex chromosomes[38], dramatic long-term reductions of neo-Y and neo-X diversity relative to the anc-X and the autosomes suggests that natural selection has been particularly intense[40] during the ongoing evolution of the mountain pine beetle neo-sex chromosomes.

Given the potential for stronger genetic drift and more intense linked natural selection acting on the sex chromosomes, we decided to conduct more thorough population-level analyses of the X and Y chromosomes. We performed reduced representation sequencing (RADseq) of an additional 30 beetles from the neighbouring OR and ID populations (15 per population, 21× per beetle), which showed very low autosomal differentiation and extensive admixture in our range-wide whole genome sequencing (Fig. 2c). Dispersal in the mountain pine beetle is high and not strongly sex-biased[41, 42]; therefore, we would only expect the persistence of strong X and/or Y differentiation over this fine geographic scale if HMS involves sex-linked interactions. Consistent with this, only the sex chromosomes clearly partitioned genetic variation between these neighbouring populations (Fig. 3a, b). Differentiation was very low for the autosomes ($F_{st} = 0.03$), elevated on the anc-X (0.15), and even higher for the neo-X (0.51) (Fig. 3c). The neo-Y again showed complete fixation ($F_{st} = 0.97$; Fig. 3a, c) of the alternative haplotypes identified using whole genome resequencing (Fig. 2d).

Analyses of population structure in OR and ID revealed further important differences between the anc-X and neo-X in individual ancestry. While the neo-X showed a clear split between OR and

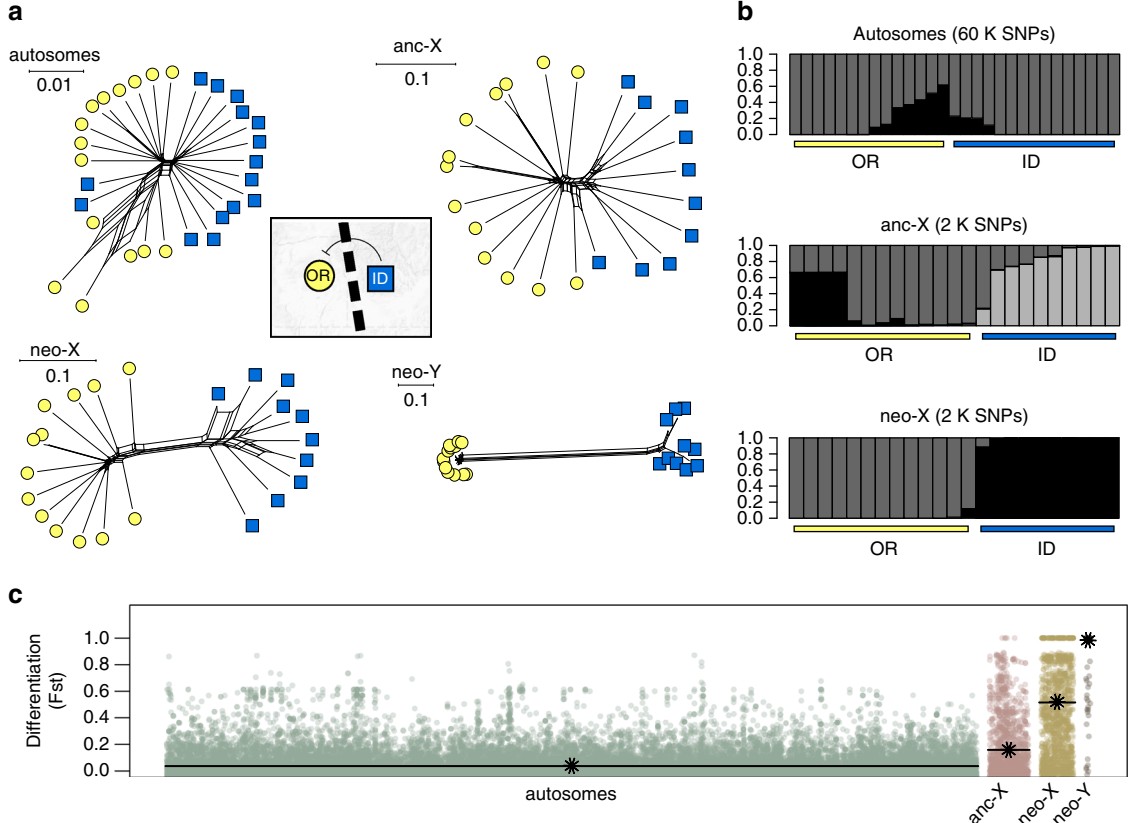

**Fig. 3** Genomic variation near the boundary of reproductive isolation. **a** Phylogenetic networks (autosomes = 40,195 SNPs; anc-X = 1853; neo-X = 1525; neo-Y = 203) from populations closest to the boundary of reproductive isolation (OR and ID). **b** Results from STRUCTURE analyses of SNPs from three linkage categories. Autosomal analyses include both males and females while anc-X and neo-X analyses included just males. Shown is the posterior probability of assignment to a cluster (K) and K = 2 for the autosomes (K = 1 was best supported), and what was identified as the best K for the anc-X (3) and neo-X (2). **c** Pairwise genetic differentiation ($F_{st}$) along concatenated scaffolds for each partition (bar and asterisk = global weighted means)

ID populations with only two individuals showing low levels of mixed ancestry, several ID beetles showed substantial OR ancestry on the anc-X (Fig. 3b). This asymmetric pattern of X-linked introgression is generally consistent with the architecture of HMS between these populations (i.e., HMS occurs in hybrid males with an ID X and an OR Y). Moreover, differential introgression of anc-X genetic markers suggests that reproductive isolation may be more strongly linked to the neo-X portion of the X chromosome. Intra-chromosomal variation in gene flow across secondary contact zones has been used to map hybrid incompatibilities in other systems[43] and our results indicate that fine-scale resolution of X-linked reproductive isolation in mountain pine beetles may be possible through expanded population sampling and a more resolved X-linked genetic map.

Collectively, these results indicate a period of allopatry leading to genetic differentiation and functional divergence between some populations followed by secondary contact and gene flow near the centre of the mountain pine beetle range. Consistent with this model, multiple host tree species show broadly concordant patterns of population isolation during Pleistocene range retractions and secondary contact within the same geographic region[44]. For example, pollen and molecular data show that several *Pinus* host trees were isolated in distinct coastal and Rocky Mountain refugia during the Pleistocene and subsequently expanded northward as the climate warmed[44].

Our findings parallel recent cases where species boundaries persist at only a subset of the genome[45]. Asymmetric hybrid incompatibilities (Fig. 2a) often reflect negative epistatic interactions that include sex-linked loci[3, 46]. Reproductive barriers are

expected to quickly break down in the face of gene flow under a classic epistatic model of hybrid incompatibilities, but may persist within co-adapted genetic pathways[47] and/or in subsets of the genome harbouring many linked incompatibilities[48]. In mountain pine beetles, sex-linked reproductive incompatibilities persist between these nascent species despite extensive autosomal admixture (Fig. 2c), but the overall strength of isolation may have partially attenuated due to introgression resulting in a distinct ring-like pattern of reproductive isolation around the Great Basin Desert (Fig. 2a). This model of allopatric population divergence followed by secondary contact and gene flow stands as a viable model for other species that present a geographic continuum of reproductive isolation. Given the rapid pace of neo-sex chromosome evolution, it also is possible that the complex geographic progression of reproductive isolation in this system (Fig. 2a) also partially reflects the ongoing accumulation of hybrid incompatibilities across parts of the mountain pine beetle range. Differentiating between these scenarios will require a much more detailed understanding of how both the genetic basis and architecture of HMS and HMI varies by geography.

**Neo-sex chromosome structural and functional evolution.** The genetic architecture of HMS combined with patterns of population genomic variation, differentiation, and admixture suggests a central role for neo-Y evolution in reproductive isolation. Male-limited sterility and limitations of the mountain pine beetle system restrict our ability to further dissect incompatibilities on the non-recombinant neo-Y chromosome. Moreover, as with most

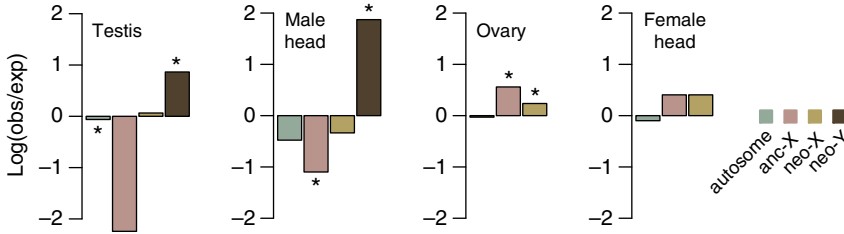

**Fig. 4** Tissue expression enrichment across the genome. Relative enrichment of genes detected only in the focal tissue (observed vs. expected gene counts detected at ≥1 fragments per kilobase per million reads mapped or FPKM, see Supplementary Table 5). Significant enrichment (positive values) or depletion (negative values) of gene counts is based on a hypergeometric test (*$p < 0.05$)

population genomic analyses of speciation, these proposed connections to reproductive isolation are indirect and removed from the molecular evolution of the neo-sex chromosomes and the functional underpinnings of relevant hybrid phenotypes. Neo-sex chromosome evolution can result in rapid functional specialization and, under some conditions, the structural degeneration of the sex-limited (Y) chromosome[15]. Here we are particularly interested in understanding if these long-term dynamic genomic changes occur on a timescale that is relevant to the evolution of male sterility between populations.

We first tested if sex-specific functional differences have evolved on the neo-sex chromosomes. Established X chromosomes tend to be enriched for female-specific functions (i.e., feminized) in insects[49] and Y chromosomes are masculinized[50], but the evolutionary tempo of this specialization remains unclear[15]. We characterized genome-wide expression patterns of multiple tissues (male head, female head, ovary, and testis) from beetles collected in Montana (i.e., East beetles). The anc-X was enriched for ovarian expression (i.e., feminized) and depleted of testis-specific genes (i.e., de-masculinized; Fig. 4 and Supplementary Table 5 and Supplementary Fig. 8). In contrast, the neo-X was slightly enriched for ovary-specific expression, but undifferentiated from the autosomes in testis-specific expression (Fig. 4 and Supplementary Table 5 and Supplementary Fig. 8) and in expression of genes detected in both reproductive tissues (Supplementary Fig. 8). The neo-Y was highly enriched for male-specific genes (Fig. 4 and Supplementary Table 5). Therefore, the anc-X and neo-Y are highly enriched for sex-specific functions but the neo-X appears to still be in the early stages of feminization. This intermediate stage of gene specialization, combined with sex-linked signatures of recurrent natural selection (Table 2), suggests that functional divergence is still actively evolving on the neo-sex chromosomes. Relatively few studies have evaluated sex-biased functional specialization of genes on neo-sex chromosomes at these early stages. Our results parallel the extensively studied *D. miranda* system, where the neo-Y is strongly masculinized and the neo-X is not yet female-biased[15] but shows accelerated adaptive evolution based on patterns of nucleotide diversity[40].

Given the overall pattern of neo-Y gene specialization (Fig. 4) and degeneration (Table 1), we reasoned that reproductively isolated West vs. East (and Southeast) populations would show elevated divergence in neo-Y chromosome structure and/or gene content. Consistent with this, we detected geographic variation in Y-linked sequencing coverage indicative of substantial structural variation between populations of beetles. Several neo-Y scaffolds showed little to no coverage in East beetles as well as some additional coverage variation restricted to the Southeast (Fig. 5a). These deletions were restricted to the neo-Y; only a few X and autosomal scaffolds showed minor differences in coverage (Supplementary Fig. 9). In total, we identified ~1 Mb of neo-Y sequence with significantly different coverage across the HMS

boundary (Supplementary Fig. 10) due primarily to large deletions in West beetles. These deletions were also fixed in our finer scale sampling of beetles from the OR and ID populations (Supplementary Fig. 11). We also detected several deletions that were private to the Southeast population (Fig. 5a), consistent with the occurrence of three divergent Y haplotypes (Fig. 2d). Full enumeration of the number of deletions was not possible given the current genome assembly, but the distribution of full and partial deletions across many scaffolds suggests widespread neo-Y degeneration through several independent mutational events. The rate of neo-Y degeneration should depend on the rate and extent to which recombination is lost between the neo-XY pair[15, 16]. *Dendroctonus* and many other beetles appear to have fully non-recombinant sex chromosomes[12], thus neo-sex chromosomes may be immediately predisposed to the rapid evolution of heteromorphy in this system and could account for the apparently rapid pace of structural degeneration.

Widespread specialization of male-specific genes on the neo-Y combined with extensive insertion-deletion variation suggests that crosses between populations with different neo-Y haplotypes may yield hybrid males that are missing genes that are essential to spermatogenesis. We observed HMS in hybrid males with West fathers, which are therefore missing ~1 Mb of the East neo-Y chromosome. We identified 11 predicted genes[29] within these deletions that were present in both East and Southeast beetles, five of which were highly expressed in testis or were male-specific in our expression analyses (Fig. 5b). One particularly strong candidate HMS gene was the peptidase family M2 Angiotensin converting enzyme (M2-ACE), which belongs to a fertility-essential gene family[51]. We also observed six putative neo-Y gene deletions between East and Southeastern beetles, and four deletions shared by the West and Southeast haplotypes (Supplementary Table 6). Only two of these genes were at least moderately expressed in male tissues and therefore these deletions may have less impact on male fertility (Supplementary Table 6). Nevertheless, Southeast neo-Y deletions may have phenotypic consequences in hybrid males that have yet to be uncovered.

Overall, gene deletions of highly expressed male-specific genes are likely to have functional consequences in hybrid males, providing an important conceptual link between the evolution of HMS and neo-XY evolution. But deletions cannot entirely explain the range of HMS that we observed. Loss of essential genes should also incur strong fertility defects within West beetles in the absence of compensatory evolution. Furthermore, additional negative epistatic interactions must be occurring in hybrid males since the neo-Y deletions are largely fixed across West populations that vary in HMS strength (Fig. 2a). The neo-X is the most likely location for interacting hybrid incompatibilities given patterns of gene flow near the contact zone (Fig. 3), but autosomal interactions also cannot be ruled out. Many homologous neo-XY gene pairs have likely been retained since these chromosomes are in the early stages of becoming heteromorphic,

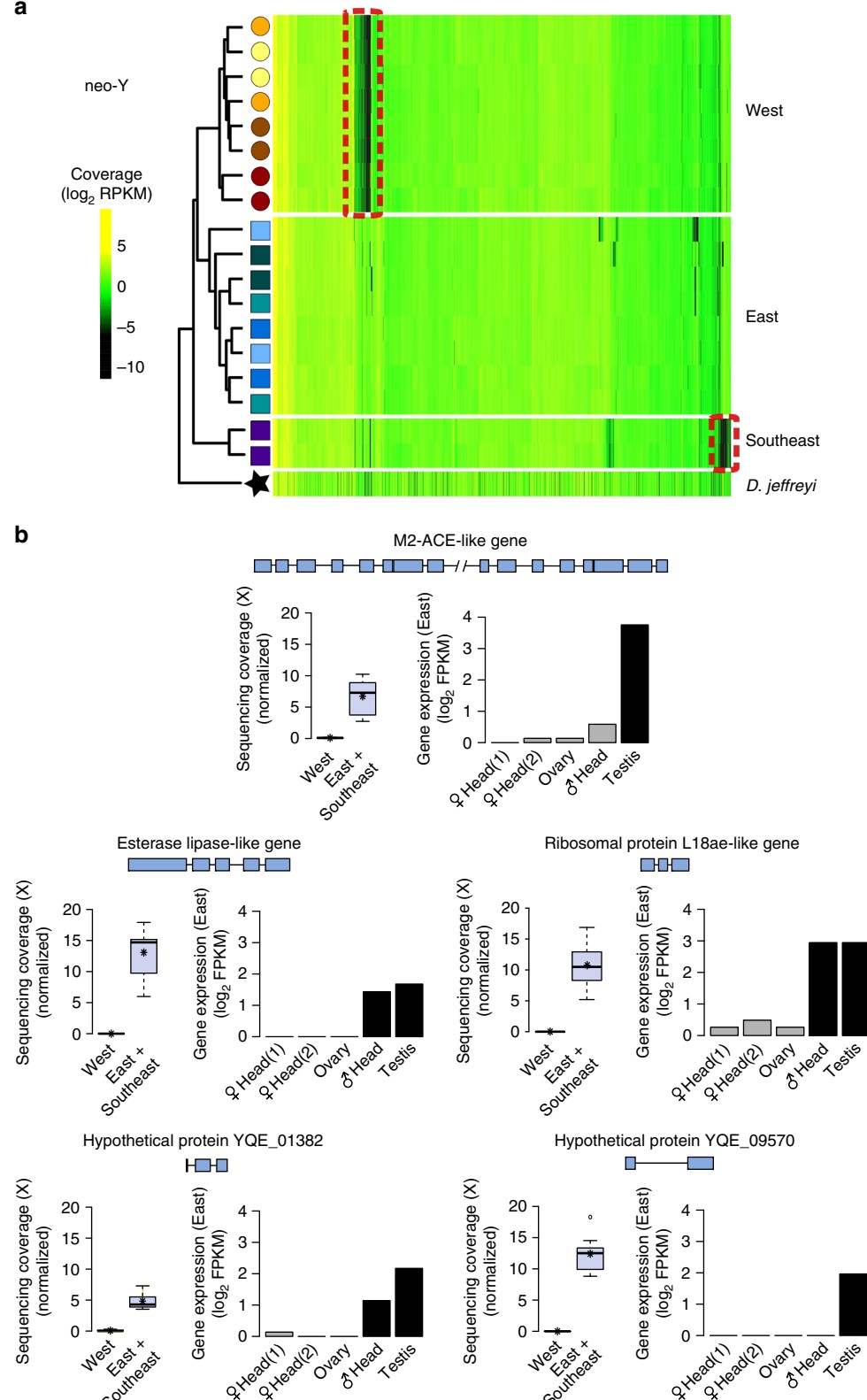

**Fig. 5** Large-scale neo-Y deletions. **a** Clustered heatmaps of sequencing coverage of neo-Y scaffolds (columns) in individual males (rows). See Supplementary Fig. 9 for X and autosome partitions. Population symbols are as in Fig. 2a and clustered deletions indicated with red boxes. **b** At least five genes were located within neo-Y deletions in West beetles. Normalized whole genome sequencing coverage and tissue expression profile in East beetles are shown (*black bars* ≥ 1 FPKM). See also Supplementary Table 6

including the putative neo-X M2-ACE homolog. Independent functional divergence of homologous neo-XY gene copies between allopatric populations could facilitate both neo-Y gene loss and the accumulation of incompatible hybrid interactions with other genomic partitions.

## Discussion

Our results suggest that sex chromosome-autosome fusions can initiate rapid divergence between populations and that this divergence is likely to have fitness consequences in hybrids. Gene movement on and off of the sex chromosomes can play an important role in speciation[52, 53] and neo-sex chromosomes are known to rapidly evolve dramatic differences in chromosome structure, gene content, and expression[15]. Given these dynamics, it seems inevitable that genomic specialization and reorganization following a sex chromosome-autosome fusion would proceed along unique trajectories in allopatric populations and upon secondary contact, hybrid incompatibilities would emerge. However, few studies have examined the dynamics of neo-sex chromosome specialization and degeneration within and between natural populations[50]. Our results indicate that neo-XY gene specialization and neo-Y structural divergence can rapidly evolve along independent trajectories between populations. This process of rapid neo-sex chromosome evolution should be enriched within reproductive pathways (e.g., spermatogenesis), which could further facilitate the persistence of reproductive barriers in the face of gene flow[47]. Here we have focused primarily on male fertility, but the observation of delayed development in some crosses raises the intriguing possibility that rapid neo-sex chromosome evolution impacts a broader range of developmental processes.

Although we observed the most striking patterns of population genetic and functional differentiation on the neo-sex chromosomes, our results do not rule out important contributions of the ancestral X chromosome to reproductive isolation in mountain pine beetles. Indeed, neo-sex chromosome systems derived from recent sex-autosomal fusions are particularly intriguing in that they are potentially subject to a broad range of processes (specialization, degeneration, meiotic drive, faster-X, dominance theory) that could contribute to the evolution of hybrid male incompatibilities (Fig. 1). Given this, functional divergence during this crucial non-equilibrium stage of sex chromosome evolution is likely to be much more rapid when compared to the evolution of established heteromorphic sex chromosomes (Fig. 1). However, these major transitions are likely to span millions of years[15, 54], and therefore could be a recurrent driver of population divergence and speciation.

That a rapid neo-sex chromosome speciation event appears to be unfolding in a phytophagous beetle irrespective of typical ecological drivers of divergence is all the more intriguing. Speciation in plant feeding insects is thought to be largely driven by host plant use[55] and beetles are by far the most diverse group of insects. Neo-sex chromosome transitions occur repeatedly across the beetle radiation[12], providing ample opportunities for this non-ecological mode of speciation.

## Methods

**Beetle collection.** Previous crosses have established asymmetric HMS between Oregon (OR) males and Idaho (ID) females, while more geographically distant crosses showed symmetric HMS[24]. Here we add new data from populations of an intermediate geographic distance (MT, nCA) and the most genetically differentiated populations (AZ, sCA[22]). For intermediate populations, we collected beetles from ponderosa pine (Pinus ponderosa) in the Lubrecht Experimental Forest, MT (46° 53′ N, 113° 27′ W) and from a lodgepole pine (Pinus contorta) from Lassen NF, CA (40° 37′ N, 121° 33′ W) in the same vicinity of the CA3 population used in Bracewell et al.[24]. For the most genetically differentiated populations, we collected beetles from the Pinaleño Mtns, AZ (32° 42′ N, 109° 55′

W), from Southwestern white pine (Pinus strobiformis) and from near Big Bear Lake, CA (34° 15′ N, 116° 54′ W), from singleleaf pinyon (Pinus monophylla).

**Experimental crosses.** Following established husbandry protocols[24], we conducted crosses between nCA and MT in 3 lodgepole pine bolts (tree sections) per cross (6 total) with 12–14 matings per bolt (36–42 male/female pairs per direction of cross). The resulting inter- and intra-population offspring were collected and pooled by cross type. We tested the reproductive capacity of hybrids with reciprocal backcrosses to both parental populations (reciprocal nCA×MT crosses ×2 hybrid sexes ×2 parent populations for backcross = 8 total). Mating pairs were replicated 18 times per combination and were randomly inserted in a 6.1-cm-wide strip of tree phloem that was caged to collect re-emerging parents. After 26 days, each strip was examined for eggs, the length of gallery (cm), and total number of eggs hatched.

We investigated both fertility and developmental timing of hybrids in the AZ×sCA cross. Previous studies have explored development time of hybrids (Supplementary Fig. 1) and developmental timing of the mountain pine beetle is likely under strong selection[56]. Experimental conditions were as above with a few minor modifications: all crosses were conducted in 4 (interpopulation) or 5 bolts (intrapopulation) per cross type with 7 matings per bolt. The sex and date of emergence was recorded for all offspring, which were then investigated for sterility by reciprocal backcrossing as above (14–35 replicates per each of 10 combinations).

**Statistical analyses of crossing experiments.** We analysed the crossing data using a mixture of generalized linear models (GLMs) and Bayesian GLMs in R (version 3.1.2). All post hoc pairwise comparisons were done with Tukey's honestly significant difference (HSD) tests using the multcomp package. We excluded crosses with <10 cm of parent gallery, resulting in analysis of 129 nCA×MT crosses (90%) and 235 sCA×AZ crosses (79%). Total egg hatch was overdispersed count data with complete separation of some levels and was therefore modelled with a Bayesian GLM and quasipoisson error distribution[57]. Gallery length was normally distributed and modelled with a GLM and Gaussian error distribution. Residual deviance was checked to assess model fit in GLMs. Results of post hoc HSD tests from these analyses are shown in Supplementary Table 1. To test for differences in the proportion of fertile male offspring among crosses, we conducted pairwise proportion tests using the pairwise.prop.test function in R with a Bonferroni correction (Supplementary Fig. 2). For the sCA×AZ cross, we tested for differences in development time using Kruskal–Wallis rank-sum tests (Supplementary Fig. 4).

**Phenotypic analyses of hybrid male sterility.** To assess hybrid male reproduction in more detail we first quantified sperm transfer to females. On their collection day, each female had her spermathecal pump and sac removed and transferred to a drop of distilled water on a microscope slide. Each structure was immediately scored under ×400 magnification on a scale of 0 (no sperm) to 4 (sperm immobilized and structure universally opaque). Differences in the sperm quantities were determined using Wilcoxon rank sum tests of all pairwise comparisons with a Bonferroni correction (Supplementary Fig. 3).

Next we examined sperm production in MT♀×nCA♂ hybrid males (22 crosses) collected during the peak of emergence from rearing containers. Hybrid males (n = 5–7 per pairing) and nCA males (n = 20) were dissected in 60 μl of phosphate buffered saline (PBS) and each testis, attached seminal vesicle, and accessory glands were isolated and transferred to 20 μl of PBS. Spermatozoa were released from the seminal vesicle into solution using fine dissection tools and 5 μl of homogenized solution was then transferred to a Makler Counting Chamber (Sefi-Medical Instruments, Ltd.). Sperm lying ≥50% within the counting grid were counted, averaging counts across the two seminal vesicles for each male. Differences between hybrid males and intrapopulation males were determined using a Wilcoxon rank sum test (Supplementary Fig. 3).

**Pheromone component production.** Pheromone samples were obtained from adult beetles collected near OR and ID (44° 16′ N, 118° 24′ W and 44° 22′ N, 115° 23′ W). Brood adults newly-emerged from logs cut from naturally-infested lodgepole pine were induced to mine into a freshly-cut lodgepole pine log in the laboratory. Solitary females (the gallery-initiating sex) that had been mining <1 d and males that had been allowed to pair 1 d earlier in the entrance of a female from the same site were excised from the bark and held in chemical adsorbent-containing microvials for 1 d at 22 °C[58, 59] to sample volatiles released by the live beetles. These steps were replicated twice. Compounds previously reported as having behavioural activity with Dendroctonus[60, 61] were identified in the hexane extracts of the adsorbent by gas chromatography-mass spectrometry by using matches of retention times and mass spectra with identified standards; compounds were quantified against a dilution curve of standards. Quantities were normalized as the percentage of the sum of all compounds found in either sex, transformed by arcsin square root, and subjected to a principal component analysis (PCA) utilizing a covariance matrix. Compounds detected in no more than trace amounts in a sex were excluded from the PCA for that sex.

**Identification of autosome and sex-linked scaffolds.** We first eliminated mountain pine beetle scaffolds that were short (<2 Kb including N's) or were likely

bacterial (identified using BLASTn against the NCBI nt database), resulting in 4,877 scaffolds (195,619,274 bp excluding N's) for further inquiry. We then took advantage of the large amount of raw sequencing data available from the male and female genome builds[29] (accessions SRX180259, SRX180261, and SRX180262) to identify putative X, Y and autosomal scaffolds using relative sequencing coverage between the sexes. We used SeqyClean version 1.8.10 to quality filter, adapter trim, and screen for common *Acinetobacter* contaminants (accessions CP000521, ACPN01000000). Filtered reads were then mapped to the reference male genome[29] using BWA-MEM version 0.7.9[62] and PCR duplicates were removed using rmdup within samtools version 1.1[63].

We counted the number of male and female reads that mapped uniquely (quality ≥20) to the male genome build scaffolds. We then used edgeR[64] to test for significant per sex, per scaffold differences in coverage by performing exact tests with a false discovery rate of 5% for each of the 4,877 scaffolds. As a proof-of-principle, we verified that six previously identified ancestral X-linked scaffolds[29] showed significant female-biased sequencing coverage (2.06 ± 0.03 fold more female coverage; Supplementary Fig. 6). We then binned scaffolds as sex-linked based on a minimum of ~1.8-fold excess coverage in females (anc+neo-X) or males (neo-Y), autosomal (<1.4-fold excess coverage in either sex), or unclassified (Supplementary Fig. 6). This procedure resulted in 1,178 autosomal, 664 anc+neo X, 2,272 neo-Y, and 763 unclassified scaffolds. Note that residual neo-XY sequence homology is expected to lead to some ambiguous or mosaic neo-X/neo-Y scaffolds in the mountain pine beetle reference genome. These complications are common to neo-sex chromosome studies but should not compromise our functional and population genomic analyses as we took additional steps to exclude potential problem regions from subsequent analyses (described below).

**Whole genome re-sequencing**. We Illumina 100 bp paired-end (PE) re-sequenced whole genomes of two adult males and one female from each of nine populations of mountain pine beetle (Fig. 2a) and one male and female Jeffrey pine beetle (*Dendroctonus jeffreyi*) (34° 13′ N, 116° 48′ W). DNA was extracted using either OMEGA E.Z.N.A Tissue DNA kits or Qiagen DNeasy kits. Sequencing libraries were prepared for fourteen samples from seven populations (1 male, 1 female; sCA, mCA, nCA, OR, ID, MT and AZ,) at the University of Montana using 300 ng of genomic DNA and the NEXTflex DNA Sequencing Kit, and DNA Barcodes by Bioo Scientific (Austin, TX). These libraries were sequenced at the Vincent J. Coates Genomics Sequencing Laboratory, Berkeley, CA. We sequenced an additional fifteen beetles including one male from each population above and two males and one female from two additional populations (UT, CO). These libraries were prepared by GENEWIZ and sequenced using 150 ng of genomic DNA, the New England Biolabs NEBNext Ultra DNA Library Prep Kit.

**Whole genome analyses**. All whole genome re-sequencing data was cleaned and mapped as above. We used the GATK Best Practices pipeline[65] to call genotypes. All BAMs were processed using realignerTargetCreator to identify regions with potential insertion-deletion variation, locally realigned using indelrealigner, and genotyped using the GATK UnifiedGenotyper version 3.1–1. We then filtered SNPs with excess depth (>60×), eliminated all indels, sites flagged other than PASS, non-biallelic positions, and individual genotypes with a genotype quality score <30. The CAN male and female samples were previously sequenced to very high coverage[29], resulting in universally high genotype qualities. To account for this high coverage bias, we only called CAN genotypes at sites that passed filters in all other individuals.

Neo-XY systems present a number of genotyping challenges related to differences in sequence complexity and residual sequence similarity between homologous X and Y-specific regions. With these issues in mind, we conducted both site and interval-based analyses using a series of conservative filters to reliably identify SNP positions across all partitions. For site-based analyses, we excluded autosomal or unclassified positions where 14 (48%) or more individuals were missing confident genotype calls (i.e., did not pass filters described above) resulting in 5,300,757 autosomal and 164,080 unclassified SNPs. To identify high quality neo-Y SNPs, we also filtered sites with confident genotypes called in any female, as well as sites called as heterozygote for any male. Both conditions should not occur on the neo-Y in males unless there are errors in the assembly and/or in read mapping. This rather conservative filtering resulted in us identifying 544 high quality SNPs (of ~60k initially identified) on the neo-Y distributed across 234 scaffolds (scaffolds constitute ~16% of the total neo-Y). The neo-sex chromosomes were not repeat-masked in advance and a large portion of filtered SNPs occurred in repetitive regions or stretches of low sequence complexity based on variance in local sequencing coverage and lower mapping qualities. To identify X-linked SNPs, we restricted analyses to males (*n* = 19) and eliminated confidently called heterozygous positions. We further partitioned the X chromosome into ancestral-X (anc-X) based on previously reported synteny with the *Tribolium castaneum* X chromosome[29] and treated any remaining scaffolds as putative neo-X. After these filters, there were 87,482 SNPs located on anc-X scaffolds and 75,450 SNPs on neo-X scaffolds. Retained SNPs had similar mapping qualities across partitions (mean MAPQ of 58.5, 59.5, 57.2, and 56.9 for autosomal, anc-X, neo-X, and neo-Y partitions, respectively).

We used PCA to test for genetic structure among individuals as implemented with Eigensoft[66]. For the autosomal PCA, all 29 mountain pine beetles were

included in the analysis, while we restricted our analyses to males for the anc-X, neo-X, and neo-Y comparisons (Supplementary Fig. 7). We further assessed genetic structure using STRUCTURE version 2.3.2.1[67] and randomly sub-sampling SNPs (50,000 generations, 20,000 generation burnin). We tested *K*-values of 1–9 and ran 6 replicates per each *K* under an admixture model. We analysed 60,000 autosomal, 15,000 neo-X, and 25,000 anc-X SNPs with X-linked analyses using a haploid model (PLOIDY = 1). The best *K* for each chromosome category was determined using STRUCTURE HARVESTER[68, 69] (Supplementary Fig. 7). We also tested for admixture using the three populations test[34] as implemented in Treemix version 1.12[70]. We tested all 360 three-way population comparisons using 1,269,066 autosomal SNPs where all individuals had a confidently called genotype, which were analysed in 1,269 blocks of 1,000 SNPs. We further explored gene flow on the sex chromosomes using the D-statistic[36] which was implemented using a block jackknife approach in the R package evobiR. We tested the populations closest to the contact zone (OR and ID) and a *Z*-score value of 3 and above considered significant[33]. Phylogenetic NeighborNet trees were generated using SplitsTree4[71] (Fig. 2d). We quantified relative genetic differentiation for the autosomes (excluding CAN) using Weir and Cockerham weighted $F_{st}$[72] estimated with VCFtools version 0.1.12b.

Next we used interval-based analyses to evaluate diversity and divergence between West and East beetles across all genomic partitions. To accurately identify both variant and invariant positions, we re-analysed and re-filtered our whole genome data. For the anc-X, neo-X and autosomes, we first restricted analyses to high quality intervals ≥100 bp in length where all individuals had coverage ≥3x and ≤1 SD of their total estimated mean coverage. The purpose of these thresholds was to ensure that all individuals had adequate coverage to call a variant but not excessive coverage as would be expected if the interval was a collapsed repeat or originated from some other assembly error. We used the same absolute coverage thresholds (≥3×) for neo-Y intervals but also eliminated regions with female coverage. We then eliminated X and Y intervals that included sites where more than one male was confidently called a heterozygote (GQ ≥20) as these are likely repetitive regions or collapsed portions of the neo-X/neo-Y. This series of step-wise filters allowed us to find regions of the degenerate neo-Y where we felt confident that we were estimating our parameters over single copy regions that were free of assembly mistakes that plague mountain pine beetle neo-Y scaffolds. Mapping qualities were similar across filtered partitions (mean MAPQ of 54.1, 59.5, 57.3 and 54.6 for autosomal, anc-X, neo-X, and neo-Y partitions, respectively). We genotyped either males or females using GATK's UnifiedGenotyper, filtered as above, and estimated nucleotide diversity and $D_{xy}$ across concatenated intervals using a custom script and generated 95% bootstrapped confidence intervals (999 replicates). Relative node depth was calculated following Feder et al.[39]. To characterize the sex-linked genetic diversity relative to the autosomes, we first estimated divergence ($D_{xy}$) to the outgroup, *D. jeffreyi*. After accounting for mutational differences (diversity/divergence to outgroup) we calculated the sex chromosome/autosomal ratio[38].

Stringent site and interval-based genotype filters were used to exclude genotyping errors given the highly fragmented and repetitive nature of the neo-Y in the genome assembly and the potential for residual X-Y sequence similarity. We further evaluated the potential impact of our site-based filters on our biological inferences in two ways. First, we removed the male-specific filters and we repeated select analyses of population differentiation ($F_{st}$), PCA and phylogenetic trees to verify that qualitative patterns between partitions were consistent. Second, we repeated X-linked population genetic analyses on females (excluding males) using autosomal filtering. All qualitative patterns were consistent across partitions under these conditions. For example, female estimates of diversity on the anc-X (East = 0.33%, West = 0.34%) and the neo-X (East = 0.17%, West = 0.29%) were similar to male-based estimates (Table 2), confirming that our results were not driven by SNP ascertainment issues in males due to lower coverage or from male-specific filtering issues. These filtering issues were further explored in our expanded RADseq analysis of the contact zone (see below).

**MtDNA assembly and analysis**. We used Velvet 1.2.10[73] to de novo assemble (kmer 75–85, −cov_cutoff = 100, −exp_cov = 1000) and MITOS (http://mitos.bioinf.uni-leipzig.de/help.py) to annotate mtDNA genomes. We then genotyped all 31 individuals using the highest quality de novo mtDNA genome assembly as a reference. Using GATK UnifiedGenotyper, we called the most frequently encountered genotype at all sites to avoid errors originating from nuclear copies of mtDNA. We confidently assembled 14,783 bp of the mtDNA genome for each beetle, including 13 protein coding genes, 2 rRNA genes, and 19 tRNAs. MtDNA sequence divergence between the mountain pine beetle and the Jeffrey pine beetle was estimated to be 6.8% and we used the insect mtDNA molecular clock estimate from Papadopoulou et al.[30].

**Identifying neo-Y linked genes and their neo-X gametologs**. We first identified all annotated genes that fell on neo-Y scaffolds (above, 781 genes total). We then used this list along with Ensembl and BioMart[74] to identify 'paralogous' sequences within the draft mountain pine beetle genome. We restricted our set of genes to those where a single clear 'paralog' was present on a scaffold not identified as neo-Y and where at least 60% of the neo-Y gene could be aligned. Genes below this threshold were too fragmented to accurately align and were often only partial hits

of dubious homology. We then downloaded coding domain sequences and aligned the putative neo-X and neo-Y gametologs using MAFFT[75] and calculated $K_s$ on 119 unambiguous neo-X/neo-Y homologs (1:1 XY pairs as defined in Supplementary Fig. 6) using KaKs_Calculator[76].

**Whole genome insertion-deletion variation.** We counted the number of reads mapping (minimum mapping quality 20) to the 4,114 scaffolds. We then used edgeR to normalize and estimate RPKM (reads per kilobase of scaffold per million reads mapped) for each individual and to test for differences in per scaffold coverage between East, West, and Southeast beetles. We used the pheatmap package in R for clustering (hclust), restricting analysis to scaffolds with a minimum of two counts per million for at least eight individuals.

**RADseq population genomic data generation and analysis.** Approximately 400 ng total DNA from adult OR and ID beetles was used for the RADseq protocol. We developed a customized single restriction enzyme digest protocol similar to Etter and colleagues[77], with additional DNA cleaning steps and a slightly modified PE2 adapter (available upon request). DNA was cleaned, digested with *PstI* (New England Biolabs), barcoded adapters were ligated to the restriction cut sites, pooled across samples and sonicated using a Bioruptor (Diagenode, Inc.). Sheared DNA was blunt-end repaired, a 5′ A was ligated to the fragment, and then a small Y-adapter was ligated to the A overhang. The indexing read barcode and Illumina specific adapter sequences were added to the fragment through PCR (5 separate 20 μl reactions: 98 °C for 30 s, 14× (98 °C for 10 s, 65 °C for 30 s, 72 °C for 30 s) and a final extension at 72 °C for 5 min). Libraries were pooled and 100 bp PE Illumina sequenced at the University of Utah Microarray and Genomic Analysis Core Facility, Salt Lake City, UT, or the Vincent J. Coates Genomics Sequencing Laboratory, Berkeley, CA.

Raw sequencing reads were processed and cleaned using the process_radtags program from Stacks version 1.18[78] and default settings. The resulting paired end reads were then mapped to the male reference genome using the BWA aln and sampe pipeline version 0.7.5 and filtered for PCR duplicates as above. We compared coverage of RADtags positioned within 141 putatively neo-Y deleted regions (mapping quality ≥20) using samtools[63] (Supplementary Fig. 11). We then genotyped RADtags from all OR and ID individuals as described above. SNP filtering differed slightly by linkage group because of differences in ploidy for the autosomes and sex chromosomes in males and the availability of male and female RAD data for different linkage groups. To call variants on the autosomes, we analysed both male and female beetle data and only kept genotypes with a minimum genotype quality of 30 and a minimum depth of 10 reads. We removed indels, excluded sites that failed quality filters, and restricted our analyses to bi-allelic sites in Hardy-Weinberg proportions where at least 20 of the 30 individuals were confidently genotyped. Neo-Y SNPs were identified from males using filters as above except we applied no minimum coverage filter and retained SNPs found in only one individual. We excluded positions with female genotypes (GQ ≤30) or male heterozygosity. X chromosome analyses were restricted to males and using the same filtering criterion as for the neo-Y. Population genetic analysis of RADseq data was as above, except we excluded SNPs with a minor allele frequency <0.1 when estimating Weir and Cockerham's weighted $F_{st}$. Given concerns for how filtering might influence our metrics for the sex chromosomes, we calculated $F_{st}$ in females for the X chromosome using autosomal filtering (as described above). We found $F_{st}$ to be qualitatively similar although somewhat less pronounced (auto = 0.03, anc-X = 0.10, and neo-X = 0.30). We further explored how removing all male-specific filters in males would influence these estimates and we found that again, the results were qualitatively similar, although less pronounced (auto = 0.03, anc-X = 0.14, neo-X = 0.30, neo-Y = 0.48).

**RNA-seq data generation.** Adult beetles were collected from newly attacked lodgepole pine near Marias Pass, MT (48° 17′ N, 113° 24′ W) in the fall of 2013. Tissues were dissected from live beetles in RNase-free PBS and RNA was extracted immediately using a Qiagen RNeasy kit. RNA integrity was verified using an Agilent 2100 Bioanalyzer (Agilent Technologies) and libraries were prepared using the Agilent SureSelect Strand-Specific RNA Library Kit (Santa Clara, CA). We generated 13 RNA-seq libraries, five of which consisted of pools of three individuals. These pools were of male heads, female heads (two replicates), ovaries and testes. Individual libraries were constructed from ovaries of four females and testes of four males. All libraries were initially Illumina sequenced in a truncated run (machine failure) of one lane of 100 bp PE and then re-sequenced with one additional lane at the Vincent J. Coates Genomics Sequencing Laboratory, Berkeley, CA. Analyses were performed on the combined data.

**Gene expression analysis.** All reads were quality filtered as above and mapped to the male reference genome[29] using Tophat2 version 2.0.10[79]. We used default parameters, specified fr-firststrand, and used the mountain pine beetle annotation file (13,218 gene models[29]) to assist in mapping. Because we had pooled data as well as individual data, we performed two separate analyses using Cufflinks version 2.2.1[80]. To allow for gene discovery, we used Cuffmerge to merge published mountain pine beetle transcriptome data with the transcriptomes from our tissues[80]. For pooled samples, we only estimated relative gene expression levels. The

individual data set had sufficient replication to allow tests for differentially expressed genes. In both analyses, gene expression values (FPKM, fragments per kilobase of transcript per million fragments mapped) were restricted to only those genes found in our 4,114 high-quality scaffolds. Next we used our pooled expression data to perform hypergeometric tests for tissue specific enrichment or depletion of counts of expressed genes across linkage groups. A gene was considered tissue specific if it was expressed in the focal tissue and not expressed in the three other tissues (FPKM = 0). These analyses were conducted with two different gene expression thresholds: focal tissue >1 FPKM and focal tissue >10 FPKM (Supplementary Table 5). All tests were conducted using phyper in R version 3.1.1.

**Identifying deleted gene intervals on the neo-Y.** We used edgeR (as above) to compare read counts of East+Southeast to West beetles over 781 annotated gene intervals on neo-Y linked scaffolds, 24 of which showed significant differences in coverage. We then identified a subset of 11 gene intervals that showed greater than 10-fold higher coverage in East+Southeast beetles with little or no coverage in West beetles. The remaining 13 significant gene intervals did not show complete loss of coverage in West beetles, and likely represent multicopy genes and/or genes with intronic regions harbouring regional indel variation.

The M2-ACE-like gene stood out an interesting candidate for playing a role in hybrid male sterility. To determine if this gene might have a homolog on the neo-X, we used BLASTp to search the neo-Y protein sequence (accession ENN80010) against GENBANK and found the top two hits were to a protein on a different male mountain pine beetle scaffold, Seq_1102825, (accession ENN74047) and to a single protein in the female mountain pine beetle genome (accession ERL91207). The presence of only one other ACE-like gene in both the male and female genome builds provided some evidence that these versions might be on the neo-X. We then performed PCA of SNPs located on the putative male neo-X scaffold (Seq_1102825) and found that the first principal component split individuals by sex and explained 33.6% of the variation providing more evidence of X-linkage. Investigation of gene expression for the putative neo-X version of the gene suggested that in contrast to the neo-Y version (Fig. 5b), it is widely expressed across tissues.

**Data availability.** Sequence and crossing data that support the findings of this study have been deposited at the NCBI Sequencing Read Archive, BioProject ID PRJNA306777 and at FigShare (https://doi.org/10.6084/m9.figshare.5479594.v1). Any additional data that support the findings of this study are available from the corresponding authors upon request.

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

## Acknowledgements

This research was supported by USDA AFRI NIFA (grant no. 2013–67011–21113), the Eunice Kennedy Shriver National Institute of Child Health and Human Development (R01HD073439; J.M.G.), and through instrumentation in the University of Montana Genomics Core funded by the M.J. Murdock Charitable Trust. We thank Jim Vandygriff, Greta Schen-Langenheim and Joseph Dysthe for help collecting data. We also thank the Good lab and the Evolutionary Genetics and Genomics community at the University of Montana for helpful discussions. We thank Lila Fishman, Doug Emlen, Dan Matute, and Matt Hahn for providing helpful comments on earlier drafts of this manuscript.

## Author contributions

R.R.B. and J.M.G. conceived of the study. R.R.B. conducted the MT×nCA cross, and performed all crossing and genomic analyses. B.J.B. conducted the AZ×sCA cross. B.T.S. conducted pheromone analyses. R.R.B. and J.M.G. wrote the manuscript with comments from all authors.

## Additional information

**Competing interests:** The authors declare no competing financial interests.

