## [Peer Review File · Nature Communications]

Reviewers' comments:

Reviewer #1 (Remarks to the Author):

This paper by Bracewell et al is an important and novel contribution to our understanding of the role of sex chromosomes in speciation. The authors show that evolution of a neo-sex chromosome system has contributed to the evolution of reproductive isolation between pine beetles in secondary contact. Demonstrating that the early stages of sex chromosome evolution contribute directly to speciation has rarely been done. The authors support their conclusions with thorough and rigorous analyses, representing an enormous amount of work from the large number of well-controlled crosses to the extensive and well-done genomics analyses. The paper is beautifully written. It is a rare pleasure to review a paper like this!

After careful reading, I have only a few minor comments:

1. L92: define "univoltine" for general reader.
2. Supp Fig. 5: define logCPM (x-axis label)
3. Fig. 1c: No labels on the f3 statistic plot.
4. In the main manuscript, clarify which population the published reference sequence is from.

Reviewer #2 (Remarks to the Author):

Comments on "Rapid neo-sex chromosome evolution coupled with incipient speciation in a major forest pest" by Bracewell et al. (Nat Comm)

The authors have evaluated population differentiation in the mountain pine beetle, a species with a gradient in degree of reproductive isolation between western and eastern populations. The species has a recently established neo-sex chromosome which is shared with its sister species (Jeffrey pine beetle). The study includes impressive population genomics analyses of the neo-sex chromosome and the rest of the genome, and presents data on genome-wide gene expression (with a focus on genes located in a fixed deletion in the western populations).

I am very impressed by the genomic and transcriptomic characterisation of the sex chromosome in multiple populations presented here. However, neo-sex chromosomes have been characterised before and the main novel aspect of the present study is the potential link to speciation. We certainly expect sex chromosomes to be involved in speciation according to theories going back to Haldane (1922), but there are only a few studies so far that claim to have been found support for a link involving neo-sex chromosomes, including the two papers that the authors refer to (Kitano et al. 2009 and Smith et al. 2016). Thus, additional empirical evidences are necessary for drawing general conclusions regarding this fascinating theory.

Does this paper then present such evidences? It seems no doubt that some of these populations, namely west vs east, have been developing reproductive isolation quite recently – thus, speciation seems to be ongoing. Also, there is no doubt that the sex chromosomes, and the neo-Y in particular, have diverged much more between populations than autosomes. But the main point is whether this proves that neo-sex chromosome evolution is involved in reproductive isolation. I am not as convinced about this as the authors, because they have also presented clear evidence of strong sex chromosome divergence (or even stronger) between some populations in the east (Figure 1d: SE and E pops are as diverged as E and W; Figure 3 A: both W and SE have large deletions compared to E) – importantly, these populations (SE and E) are not reproductively isolated. One can therefore draw the conclusion that rapid sex chromosome evolution coincides with reproductive isolation in one case (W vs. E) and with no reproductive isolation in a second case (SE vs. E). Thus, this study does not show that sex chromosome evolution is involved in the evolution of reproductive isolation.

Furthermore, there is a gradual pattern of reproductive isolation between western and eastern populations so that more closely located populations show weaker reproductive isolation than populations located further away. There is no corresponding gradual differentiation in the genomic differentiation as could have been expected.

This does not mean that I do not believe that genes located on the sex chromosome may be involved in speciation in general (there are plenty support for Haldane's rule in the literature) or in this particular case, but only that I do not think that the authors are able to demonstrate such link with the present data. It is a fascinating study about rapid sex chromosome evolution in populations that have and have not evolved reproductive isolation. All parts of the results should be given equal weight in the interpretation of the results, and one also becomes interested in timing (e.g. when did these populations evolve all this sequence variation and fixation in relation to the glacial periods?).

Reviewer #3 (Remarks to the Author):

Bracewell et al. carried out lab crossing experiments to characterize strength of reproductive isolation between populations of mountain pine beetle. They also characterized genome-wide genetic differentiation between these populations, aiming to identify genomic regions associated with reproductive isolation mechanisms.

Although the research system is very unique and has a great potential to provide a novel finding in the genetic mechanisms of reproductive isolation and evolutionary processes in speciation, the evidence is circumstantial and too weak to conclude that neo-XY chromosomes are involved in reproductive isolation. Gene degeneration, accumulation of transposable elements, elongation of introns, etc., on Y-chromosome is well known and an inevitable fate after the formation of neo-Y accompanied by the cessation of recombination between gametologs. This molecular evolution on Y chromosome seems to be happening in the pine beetle (despite some technical problems), but this does not necessarily mean their

involvement in reproductive isolation. What is shown is merely a correlation and not a causal relationship.

In addition, I also have a serious technical concern about SNP filtering on neo-XY chromosomes (see my specific comments below). The potential bias introduced by this filtering process can spuriously inflate genetic differentiation on these chromosomes (Fig 1C), and thus cannot be interpreted as a consequence of speciation. Given more recent reduction in N_e on neo-X than anc-X, lower diversity should be expected on anc-X than neo-X under neutral expectation. This clearly contradicts to the observed pattern (Table 1) unless there are a large number of loci on neo-X under strong selection. On the contrary, the evolution of sex specific gene expression, which can be an integral part of reproductive isolation mechanisms, is not detected on neo-X but is detected on anc-X. Thus, it is likely that higher differentiation and lower diversity on neo-X can be in part due to the technical artifact.

The research system should be introduced a bit more carefully in light of chromosomal speciation models. Previous work introduced in Introduction section (Bachtrog's and Kitano's work on *Drosophila* and stickleback) is fundamentally different from the system used in this study because their studies show that chromosomal rearrangements on sex chromosomes are associated with reproductive isolation between populations of ancestral karyotype and derived neo-XY karyotype. However, reproductive isolation mechanisms between populations of pine beetle have likely established after the fixation of the neo-XY karyotype. Thus, the authors should introduce the system not in the context of sex chromosome speciation but in the context of the 'fast-X(Y)' theory.

Finally, it is questionable how the novelty that the authors claim (Line 111-113) is relevant to the better understanding of the process and mechanisms of speciation. Increase in the strength of reproductive isolation by geographic (or more directly genetic) distance has been known for decades (eg, ring species). It is perhaps more meaningful to compare the extent of reproductive isolation and genetic distance.

Specific comments:

Line 68-78:

It is not clear how populations of mountain pine beetles are karyotypically different from each other. Line 72-73 describes the karyotype evolution between species (mountain pine beetle vs. Jeffrey pine beetle), this tells nothing about within species polymorphisms in karyotype.

Line 73: "Despite low levels of genetic differentiation..."

I would like to see the actual numbers without searching for the ref 23.

Line 86: It is not clear how the central boundary is defined.

Line 90: It is unusual to consider "delayed development" as a character of hybrid inviability. In a strict sense, no matter how development is delayed, hybrids can be perfectly viable and

fertile.

Line 91: "geographically distant cross"

I guess that an assumption for this pair of populations to be the most distant is that scattered small patches of host trees shown in grey in Fig 1a do not provide suitable habitat for the study species. In addition, the Great Basic Dessert provides a geographic barrier. This should be described at somewhere in the text.

Another point is that why geographic distance was used and not the genetic distance. I think that the underlying assumption is 'isolation-by-distance', where the most geographically distant population pairs are also genetically distant. If so, it should be presented in more direct way. This is particularly important in light of 'speciation-with-gene-flow' because positive correlation between the level of genome-wide divergence and the strength of reproductive isolation is expected.

Finally, results of HMI for less distant population pairs should also be shown in Suppl.Fig3. For the same reason explained above, it is important to show weak or no HMI for less "genetically" distant populations.

Line 92: I do not understand this argument. Development timing for bivoltine insects should also be under strong selection.

More importantly, it is very confusing to introduce development timing as a mechanism of reproductive isolation here, especially right after the result of HMI (delayed development of male hybrids). Are Line 91 and 92 in the same paragraph? It is unusual to consider temporal isolation as a form of hybrid inviability. Without showing results of HMI for OR and ID crosses, it seems pointless to discuss ecological segregation (or similarity in this case) between OR and ID.

I would suggest to present these results in a different way. First, there is little evidence for ecological segregation between OR and ID populations (and perhaps AZ and sCA, the pair shown in Suppl. Fig.3). However, AZ-sCA F1 hybrids show delayed development, which is a transgressive character (much more delayed than both parents). This could represent a strong barrier to gene flow. (Nonetheless, AZ and sCA populations appear to be geographically isolated for the first place, so it is not clear whether this temporal isolation plays any role in speciation).

Line 142-156:

Here the stringent filtering for sex chromosomes becomes problematic. Higher proportion of fixed sites on Y chromosome, for example, could be due to its smaller effective population size, stronger selection, as well as artifacts by removing 'heterozygous' sites. It is important to show how divergent X and Y gametologous regions. If they are young enough, then the vast majority of neoX and neoY is the pseudo-autosomal region (PAR), which behaves just like autosomes. Therefore, removal of het sites can induce serious bias toward fixed sites. In addition, mtDNA is essentially one marker no matter how many SNPs it has. Thus, non-concordance with the 'boundary' could be a reflection of stochasticity.

One can infer the location of PAR (or less differentiated regions) by using read coverage information.

Line 208-209:

The difference between anc-X and neo-X may reflect the mappability of X-reads and Y-reads onto the X and Y reference genomes. Since there is no gametogs for anc-X, this region is less sensitive to the filtering process. However, neo-X is heavily influenced, resulting in the higher differentiation.

Line 253-254: "These patterns are seemingly at odds with our inferences of sex-linked HMS and reduced gene flow."

Lower D_{xy} for X and Y is not surprising given the smaller N_e for sex linked markers. ($D_{xy} = 2ut + 4N_eu$ for diploid markers. See Nachman and Payseur 2012)

Line 309: "The neo-Y was highly enriched for male-specific genes (Fig. 4a and Supplementary Fig. 10)."

Supplementary Fig. 10 does not have plot for neo-Y.

Line 446: two reference genomes with 3 accessions?

Line 457-458:

Clarify how male-to-female ratio is calculated. For X chromosome female/male results in $>2x$, but not for Y if one takes female/male (and hence must be male/female??)

Line 487: "this bias"

Which bias?

Line 490: "confident genotype calls"

Please describe. Are these synonymous to the sites that passed the filter?

Line 491-496:

I am seriously concerned about potential artifacts introduced by the SNP filtering on X and Y chromosomes. It is not clear when the neo-XY chromosomes are formed and how divergent they are, but I assume that they have relatively recent origin with substantial sequence similarity. The fate of newly formed new-XY chromosomes is described elsewhere (see Bachtrog papers). Briefly, at the onset of X-autosome fusion, the whole neo-X and neo-Y are essentially the pseudo-autosomal region (PAR). As recombination is suppressed, the size of PAR reduces and differentiation increases between neo-X and neo-Y. The pine beetle system represents some intermediate stage of sex chromosome differentiation with a number of gametologous reads being mapped to the alternate sex chromosomes. Therefore, these filter sets can remove less divergent regions including PAR and choose highly differentiated regions only.

Line 501: 29? Not 27?

Line 523:

If the same coverage thresholds were used for Y with autosomes, it is in practice coverage >6x-equivalent in autosomes.

Line 526:

Male het sites on Y-chromosome also represent Y-regions that are not diverged from X-gametologous regions. Similar to the stringent filter that the authors used, this filter removes, true Y regions with lower divergence. Thus the bias introduced by their stringent filter (Line 519-520) cannot be removed.

Line 543:

There are a number of publicly available tools to detect indels and other structural variations with much more sophisticated approaches by combining read-map, split-read, and read-depth information (eg., Manta SV, Lumby, Delly, Breakdancer, etc. see Pirooznia et al. 2015 for comprehensive review). I would recommend to use some of these tools to quantify SV frequency and density across the genome and compare it between auto-x-y chromosomes.

Line 544:

What is 'high quality scaffolds'?

Line 547: "pheatmap"

Typo?

Figure 1C, Admixture

No Axis labels?

Supplementary Table 2:

This is a figure and not table.

Supplementary Figure 2A: Use larger font size. X-axis labels, in particular, are not legible. The scale in Y-axes is not described. Are these the same with the one in 2B? I guess not. Colour code (red, yellow, green) is not described.

Supplementary Figure 3:

"The dashed [vertical] line represents the median development time for each sex.". They should be colour-coded for different sexes.

Supplementary Figure 5:

Methods describes log-fold change as female/male ratio (Line 456-459), while this figure appears to show male/female. Use one of the other consistently.

Reviewers' comments:

Reviewer #1 (Remarks to the Author):

This paper by Bracewell et al is an important and novel contribution to our understanding of the role of sex chromosomes in speciation. The authors show that evolution of a neo-sex chromosome system has contributed to the evolution of reproductive isolation between pine beetles in secondary contact. Demonstrating that the early stages of sex chromosome evolution contribute directly to speciation has rarely been done. The authors support their conclusions with thorough and rigorous analyses, representing an enormous amount of work from the large number of well-controlled crosses to the extensive and well-done genomics analyses. The paper is beautifully written. It is a rare pleasure to review a paper like this!

Response: *We thank the reviewer for their enthusiastic comments. We have addressed all of their suggestions as indicated below.*

After careful reading, I have only a few minor comments:

1. L92: define “univoltine” for general reader.

Response: *Now defined as “univoltine (one generation per year)” on Line 125.*

2. Supp Fig. 5: define logCPM (x-axis label)

Response: *Done (now Supplementary Fig. 6)*

3. Fig. 1c: No labels on the f3 statistic plot.

Response: *We have now added a cardinal direction and HMS boundary designation (hash from Fig. 2a) to the plot to help make clear that this is a map. The figure caption clarifies that, “Significantly admixed populations are positioned by geography”. Hopefully this makes it clear.*

4. In the main manuscript, clarify which population the published reference sequence is from.

Response: *The reference genome is from CAN, this is now clarified in the text (line 161)*

Reviewer #2 (Remarks to the Author):

Comments on “Rapid neo-sex chromosome evolution coupled with incipient speciation in a major forest pest” by Bracewell et al. (Nat Comm)

The authors have evaluated population differentiation in the mountain pine beetle, a species with a gradient in degree of reproductive isolation between western and eastern populations. The species has a recently established neo-sex chromosome which is shared with its sister species (Jeffrey pine beetle). The study includes impressive population genomics analyses of the neo-sex chromosome and the rest of the genome, and presents data on genome-wide gene expression (with a focus on genes located in a fixed deletion in the western populations).

I am very impressed by the genomic and transcriptomic characterisation of the sex chromosome in multiple populations presented here. However, neo-sex chromosomes have been characterised before and the main novel aspect of the present study is the potential link to speciation.

We certainly expect sex chromosomes to be involved in speciation according to theories going back to Haldane (1922), but there are only a few studies so far that claim to have been found support for a link involving neo-sex chromosomes, including the two papers that the authors

refer to (Kitano et al. 2009 and Smith et al. 2016). Thus, additional empirical evidences are necessary for drawing general conclusions regarding this fascinating theory.

Response: *We thank the reviewer and agree that the connection to speciation is one of the most important contributions of our study. However, we believe that there are at least two additional considerations on the novelty of our study that warrant mentioning. First, previous works on linking neo-sex chromosome evolution to speciation have either been tenuous (Smith et al. 2016) or in reference to isolation between populations with different sex chromosome systems (i.e., the initial establishment phase of neo-sex chromosomes) as presented by Kitano et al. 2009. As explained in more detail below (see Reviewer 3), our focus is on the specialization and degeneration phase of neo-sex chromosome evolution, which is both novel and important for understanding how neo-sex chromosomes may contribute to the process of speciation. We have added a conceptual figure (now Figure 1) and edited the introduction to make this important distinction clearer (see lines 53-76). Second, though we appreciate that others have examined the early stages of neo-sex chromosome evolution, many important details remain unresolved. For example, most studies of this kind have been comparative and very few have examined neo-sex chromosome evolution from a population perspective. Crucially, no previous study has presented data linking the hallmarks of neo-Y evolution (degeneration, gene specialization) to functionally relevant differences between populations (variation in male-specific gene content). We failed to make these distinctions clear in the previous submission, and have tried to re-emphasize these links as an important contribution throughout the revised manuscript.*

Does this paper then present such evidences? It seems no doubt that some of these populations, namely west vs east, have been developing reproductive isolation quite recently – thus, speciation seems to be ongoing. Also, there is no doubt that the sex chromosomes, and the neo-Y in particular, have diverged much more between populations than autosomes. But the main point is whether this proves that neo-sex chromosome evolution is involved in reproductive isolation. I am not as convinced about this as the authors, because they have also presented clear evidence of strong sex chromosome divergence (or even stronger) between some populations in the east (Figure 1d: SE and E pops are as diverged as E and W; Figure 3 A: both W and SE have large deletions compared to E) – importantly, these populations (SE and E) are not reproductively isolated. One can therefore draw the conclusion that rapid sex chromosome evolution coincides with reproductive isolation in one case (W vs. E) and with no reproductive isolation in a second case (SE vs. E). Thus, this study does not show that sex chromosome evolution is involved in the evolution of reproductive isolation.

Response: *This is a great point and we now see how this pattern would seem to partially contradict the proposed connection between neo-Y divergence and reproductive isolation. We had largely ignored the Southeastern population in our original version to keep the presentation concise and because reproductive isolation involving this isolated AZ population has not been thoroughly characterized. We now see that this was not clear in our previous submission. We have clarified that our crossing data, though extensive, did not include all possible pairwise comparisons (see Supplemental Figure 1, lines 116-120, lines 224-230). We have now revised the manuscript to more clearly explain these limitations and give equal weight to all patterns in our data. As explained in the new Supplemental Figure 1, we know a bit less about the details of sterility involving the Southeastern population. We have preliminary data that indicate males*

from SE x E crosses are probably mostly fertile but technical issues encountered during this cross make quantitative comparisons tenuous. Given that the relative strength of sterility varies across populations, we would prefer to not include these data here.

We also now more clearly explain the overall pattern of genic differences associated with the deletions. The Southeastern neo-Y haplotype, though quite divergent in structure and sequence, only shows deletions of two moderately expressed male-specific genes relative to East populations (Supplemental Table 6) and neither appears as a strong candidate hybrid male sterility gene based on function. Therefore, based on gene content, it is not clear if we would expect hybrid male sterility between these populations. We now present the expanded gene deletion data in Supplementary Table 6 and discuss these data as follows (lines 400-405):

“We also observed six putative neo-Y gene deletions between East and Southeastern beetles, and four deletions shared by the West and Southeast haplotypes (Supplementary Table 6). Only two of these genes were found to be at least moderately expressed in male tissues and therefore these deletions may have less impact on male fertility (Supplementary Table 6). Nevertheless, Southeast neo-Y deletions may have additional phenotypic consequences in hybrid males that have yet to be uncovered.”

Furthermore, there is a gradual pattern of reproductive isolation between western and eastern populations so that more closely located populations show weaker reproductive isolation than populations located further away. There is no corresponding gradual differentiation in the genomic differentiation as could have been expected.

Response: *We agree that neo-sex differentiation corresponds generally with the occurrence of reproductive isolation between populations but does not scale with the strength of isolation. We assume that sex-linked incompatibilities are involved in many epistatic interactions, consistent with previous theoretical and empirical work on hybrid incompatibilities. As the strength of isolation does generally scale with the distance from the West-East admixture zone, we propose that this variation reflects the partial breakdown of incompatibilities due to gene flow (i.e., introgression resulting in the loss of some Dobzhansky-Muller incompatibilities). At least partial break down of epistatic incompatibilities is generally expected in the face of gene flow. This pattern could also reflect the ongoing accumulation of hybrid incompatibilities. We now present a general overview of these alternative models (lines 322-338) and with special reference to neo-Y divergence again on (lines 406-419).*

This does not mean that I do not believe that genes located on the sex chromosome may be involved in speciation in general (there are plenty support for Haldane’s rule in the literature) or in this particular case, but only that I do not think that the authors are able to demonstrate such link with the present data. It is a fascinating study about rapid sex chromosome evolution in populations that have and have not evolved reproductive isolation. All parts of the results should be given equal weight in the interpretation of the results, and one also becomes interested in timing (e.g. when did these populations evolve all this sequence variation and fixation in relation to the glacial periods?).

Response: *We thank the reviewer for these very helpful comments. We regret that aspects of our previous presentation were unclear. Hopefully these revisions now provide a more balanced and compelling interpretation of our results. With respect to timing, we also now provide some*

discussion as to the historical context of allopatry (lines 322-338) and some more data on the stage of neo-XY differentiation relative to other studies (lines 170-183).

Reviewer #3 (Remarks to the Author):

Bracewell et al. carried out lab crossing experiments to characterize strength of reproductive isolation between populations of mountain pine beetle. They also characterized genome-wide genetic differentiation between these populations, aiming to identify genomic regions associated with reproductive isolation mechanisms.

Although the research system is very unique and has a great potential to provide a novel finding in the genetic mechanisms of reproductive isolation and evolutionary processes in speciation, the evidence is circumstantial and too weak to conclude that neo-XY chromosomes are involved in reproductive isolation. Gene degeneration, accumulation of transposable elements, elongation of introns, etc., on Y-chromosome is well known and an inevitable fate after the formation of neo-Y accompanied by the cessation of recombination between gametologs. This molecular evolution on Y chromosome seems to be happening in the pine beetle (despite some technical problems), but this does not necessarily mean their involvement in reproductive isolation. What is shown is merely a correlation and not a causal relationship.

Response: *We thank the reviewer for their detailed comments, which have revealed to us a number of places where we failed to explain our data clearly or completely. We have made every effort to address all of these concerns with revisions outlined as point-by-point responses below. In particular, we have restructured the introduction, results and discussion to clearly present our novel contributions (of which we believe there are several). We also present more details that help strengthen connections between genetic patterns of reproductive isolation, population genetic differentiation, and functional evolution of the neo-sex chromosomes. Aspects of these more detailed patterns clarify and strengthen the inferred connection between reproductive isolation and neo-Y divergence (see above our response to Reviewer 2). We have also made a concerted effort to acknowledge aspects of inferences that are indirect. We fully acknowledge that, in general, inferences into the process of speciation based on genetic patterns - from phylogenies, population genetic patterns of genetic differentiation, gene flow, etc, - are necessarily indirect. However, by merging population genomic data with extensive information on phenotypes of reproductive isolation in controlled crosses and variation in male specific gene content, we believe that our study goes a great deal further than most in exploring and testing these connections. That said, we have also tried to do a better job of acknowledging the limits of our inferences and the study system.*

We want to emphasize that our study does not reflect a post hoc interpretation of unconnected experiments. Rather, we have presented our data exactly as it was collected and interpreted: A series of hypothesis driven genetic and genomic experiments to understand neo-sex chromosome evolution and reproductive isolation in this important system. We believe that this distinction matters.

- *We observed extensive reproductive isolation in experimental crosses. These patterns reveal a remarkable amount of intrinsic reproductive isolation that was not predicted from what has been known regarding the ecology or phenotypic variation in this extensively studied forest pest. The distinct asymmetric patterns of reproductive isolation*

lead to the hypothesis that sex-linked or sex-limited incompatibilities must be involved in the evolution of hybrid male sterility.

- *We then annotated sex-linked regions of the beetle genome and generated population genomic data to test for genetic patterns that are consistent with the experimentally identified patterns of reproductive isolation. From these data we can reach two important conclusions. First, the extreme patterns of intrinsic reproductive isolation are occurring on a background of very low overall genetic divergence, establishing that intrinsic hybrid incompatibilities are accumulating at an unusually rapid rate in this system. Second, we can exclude the mtDNA as plausibly playing a role in reproductive isolation. The neo-Y shows extreme haplotype differentiation that is coincident with boundaries of hybrid male sterility. This correspondence is an indirect association between reproductive isolation in the lab and population genetic variation. However, these boundaries are only predicted from the crossing data and are not observed in the autosomal data. We agree that simply identifying genetic structuring across the genome, without the careful a priori hypotheses motivated by phenotypic and geographic characterization of reproductive isolation, would be a common form of speculation in the speciation literature. That is not what we did.*
- *Given the potential issues of relative genetic differentiation (F_{st}) when considered in isolation, we then test for patterns of gene flow and detected autosomal and X-linked admixture at the boundary of reproductive isolation. This allowed us to establish a link between genetic differentiation and actual gene flow and to understand the population history of this system.*
- *Given the small range-wide sample sizes used to infer these patterns, and the associated issue of lower effective population sizes on the sex chromosomes, we next collected an independent genetic dataset (RAD-seq) and tested if the broader range-wide differentiation of the neo-Y persists with larger samples from neighboring populations on either side of the experimentally defined hybrid male sterility boundary. It did.*
- *We believe that these connections are both compelling and novel in several ways (extreme patterns of sex-linked reproductive isolation over vary shallow evolutionary divergences, extreme reductions in neo-XY diversity, complete fixation of alternative neo-Y haplotypes), but these patterns are still not directly related to neo-sex chromosome evolution per se. To do this, we then used a series of expression studies to characterize the status of neo-XY specialization. We show that this system is still in the early stages of neo-X specialization. This type of experiment has only been done in a few experimental neo-XY systems and our data provide a much needed additional data point.*
- *We then detected an extraordinary pattern of between population structural divergence (insertion-deletion variation) on the neo-Y. Though, as suggested, the mutational mechanisms and end point of heteromorphic sex chromosome evolution is theoretically predicted over millions of years, it has not at all been clear if neo-Y degeneration ever occurs on a timescale that is relevant to functional divergence within and between closely related populations. Our data establishes this connection, making an important contribution to the dynamics sex chromosome evolution.*
- *We then ask if the crosses that show hybrid sterility are also missing genes that we experimentally verified as male expressed and male specific. We don't have information for all possible cross types but crosses that show some male sterility are missing Y-linked genes, crosses that are fertile appear to have the same Y-linked gene contents.*

These are the data as they were generated and interpreted. There are limitations to the system. For example, Y-linked male sterility is very hard to analyze in a quantitative genetic design and, even if we could conduct an additional quantitative genetic experiment, a simple backcross would require at least another year of experimental work (these crosses are very labor intensive and take several months per generation to conduct). Multiple 50-100 year old live pine trees need to be harvested prior to every generation of a crossing experiment and maintaining lines more than a few generations is exceptionally difficult. We feel very comfortable with this line of reasoning, both in terms of the strength and novelty of the insights, and how our approach and data compares to the high standards of the field and this journal.

In addition, I also have a serious technical concern about SNP filtering on neo-XY chromosomes (see my specific comments below). The potential bias introduced by this filtering process can spuriously inflate genetic differentiation on these chromosomes (Fig 1C), and thus cannot be interpreted as a consequence of speciation.

Response: *We understand these concerns and thank the reviewer for pointing out the need to present more details on our SNP filtering. We viewed these issues as the biggest single challenge of our study (and any study on neo-sex chromosomes) and the final results reflect careful and in depth analyses conducted over several months after the final data were collected. We failed to convey the depth of these analyses or to take the time to convince the reader that the observed patterns are not the consequence of filtering bias. We now present the following additional details.*

- *Methods. We have revised the section on SNP filtering to better convey the central issues underlying our SNP analyses, why certain filters were applied, and how these decisions impact the results (see below).*
- *We have included additional details on differences between the autosomal and sex-linked partitions and how our SNP filtering approach impacts important metrics, such as mapping qualities (Table 1, lines 595-596, 629-630).*
- *One of the central issues is the potential impact of sex-specific filters on neo-X and Y patterns of variation and differentiation, relative to the autosomes or the ancestral X. We have added additional analyses showing that overall patterns of differentiation persist when we (i) relax or remove neo-Y and X-specific filters, and when (ii) we filter the X chromosome in females and treat them the same as the autosomes. As expected, the liberally filtered (or unfiltered) data become much noisier, but the strength of sex-linked patterns is so strong that they persist under all examined filtering strategies (lines 638-650, 710-716).*

Given more recent reduction in N_e on neo-X than anc-X, lower diversity should be expected on anc-X than neo-X under neutral expectation. This clearly contradicts to the observed pattern (Table 1) unless there are a large number of loci on neo-X under strong selection. On the contrary, the evolution of sex specific gene expression, which can be an integral part of reproductive isolation mechanisms, is not detected on neo-X but is detected on anc-X. Thus, it is likely that higher differentiation and lower diversity on neo-X can be in part due to the technical artifact.

Response: We agree that the strong reduction in variation on the neo-X and neo-Y is consistent with stronger selection on the neo-sex chromosomes. This is the argument that we presented in the previous submission and which we have further refined here (see line 262-296). These results are important to inferences here but not particularly surprising given that several other studies have shown greatly reduced sex-linked variation, for both neo (e.g., *D. Miranda*, see Bachtrog and Charlesworth 2000) and old-sex chromosome systems (e.g., Y variation in humans, see Wilson-Sayres et al 2014). We don't understand the argument that because the neo-X shows reduced variation but does not yet show female-bias, that these results must be a technical artifact. Indeed, perhaps the best-studied neo-sex chromosome system is *D. miranda*, which shows extensive evidence for intense selection on the neo-X based on similar patterns of reduced variation, selective sweeps, and protein evolution but shows even less female-biased expression than the beetle neo-X (see Bachtrog et al. 2009, Zhou and Bachtrog 2012). We have added these details to the text as follows (lines 365-371):

“This intermediate stage of gene specialization, combined with sex-linked signatures of recurrent natural selection (Table 2), suggests that functional divergence is still actively evolving on the neo-sex chromosomes. Relatively few studies have evaluated sex-biased functional specialization of genes on neo-sex chromosomes at these early stages. Our results parallel the extensively studied *D. miranda* system, where the neo-Y is strongly masculinized and the neo-X is not yet female-biased¹⁵ but shows accelerated adaptive evolution based on patterns of nucleotide diversity⁴².”

The research system should be introduced a bit more carefully in light of chromosomal speciation models. Previous work introduced in Introduction section (Bachtrog's and Kitano's work on *Drosophila* and stickleback) is fundamentally different from the system used in this study because their studies show that chromosomal rearrangements on sex chromosomes are associated with reproductive isolation between populations of ancestral karyotype and derived neo-XY karyotype. However, reproductive isolation mechanisms between populations of pine beetle have likely established after the fixation of the neo-XY karyotype. Thus, the authors should introduce the system not in the context of sex chromosome speciation but in the context of the 'fast-X(Y)' theory.

Response: We really appreciate this suggestion. Indeed, we had tried to make this distinction in the previous submission, but we were not very clear. We now explain that our focus is on the specialization and degeneration phase of neo-sex chromosome evolution, which is both novel and important for understanding how neo-sex chromosomes may contribute to the process of speciation. We view this as distinct from both the establishment phase (i.e., karyotype divergence) as well as the long-term evolutionary dynamics of highly heteromorphic sex chromosomes caused by differential exposure of recessive mutations in males (i.e., faster-X and related processes). Indeed, the remarkable variation in neo-Y gene context segregating between closely related populations of beetles is quite unlike anything one would expect to find in an 'old' sex chromosome system. We have edited the introduction to make this important distinction clearer and added a new conceptual figure to convey these processes (see lines 53-76; see Figure 1).

“It is now apparent that transitions to new (neo) sex chromosomes are common over evolutionary timescales¹⁰⁻¹², with some groups showing high levels of turnover in chromosomal systems of sex

determination^{10,11,13}. Neo-sex chromosomes can originate through diverse mechanisms, such as the *de novo* evolution of a sex determining factor on otherwise homomorphic autosomal chromosomes (e.g., wild strawberry¹⁴) or through the fusion of an ancestral sex chromosome with an autosome (e.g., *Drosophila miranda*¹⁵). Given the surprisingly fluid nature of sex chromosome systems in some groups, there are at least three general stages of sex chromosome evolution – each with distinct evolutionary dynamics – that might contribute to the process of speciation (Fig. 1). First, the establishment of a new sex chromosome system within a population could lead to reproductive isolation between species with different sex chromosome systems, as has been recently shown in threespine sticklebacks^{16,17}. Second, once established, the functional conversion of autosomes to sex chromosomes results in dynamic changes in chromosome structure, gene content, and expression underlying the evolution of sex-specific functions^{15,18,19}. The extensive genic specialization and structural degeneration accompanying such chromosomal transformations provide some of the most extreme examples of long-term genome evolution²⁰. In XY systems, the rate of sex chromosome differentiation and neo-Y degeneration is dependent on the suppression of recombination between the neo-XY pair^{15,18}. In principle, sex chromosome transitions could also result in rapid functional divergence and the accumulation of genetic incompatibilities between populations, but the tempo of these genomic changes and their contribution to broader patterns of species diversification remains largely unknown. Finally, it is well established that older, highly heteromorphic sex chromosomes often play a disproportionately large role in speciation through the exposure of recessive genetic incompatibilities in hybrid males⁴ and various evolutionary dynamics (e.g., faster-X, meiotic drive) that can drive rapid sex-linked divergence^{5,6}."

We then return to this in the conclusions, emphasizing that given the rapid but protracted timeline of neo-XY specialization, we might expect that this phase of sex chromosome evolution to play a recurrent role in population divergence and speciation (see lines 436-441)

"Functional divergence during this crucial non-equilibrium stage of sex chromosome evolution is likely to be much more rapid when compared to the evolution of established heteromorphic sex chromosomes. However, these major transitions still are likely to span millions of years^{15,56}, and therefore could be a recurrent driver of population divergence and speciation. In contrast, the establishment of new a sex chromosome system is likely to be limited to a single bifurcation event^{16,17}."

As explained below, we have also made edits throughout to more clearly quantify the extent of divergence between the neo-XY pair (see lines 175-183, lines 367-371).

Finally, it is questionable how the novelty that the authors claim (Line 111-113) is relevant to the better understanding of the process and mechanisms of speciation. Increase in the strength of reproductive isolation by geographic (or more directly genetic) distance has been known for decades (eg, ring species). It is perhaps more meaningful to compare the extent of reproductive isolation and genetic distance.

Response: *We think we are in agreement with the reviewer and that our statement was misinterpreted or unclear. To clarify: Yes, ring species have been known for decades. First, it is actually questionable whether any proposed ring species actually corresponds to the original model (i.e., as we see in beetles, most ring systems now appear to reflect historic allopatry to some degree). Second, to our knowledge, none of these species present a geographic continuum of intrinsic reproductive isolation (i.e., hybrid incompatibilities, versus variation in ecological and/or pre-mating isolation). This distinction is important as the processes underlying such forms of divergence are different and intrinsic incompatibilities are generally assumed to not evolve sufficiently fast to result in this pattern. We believe that such a broad geographic*

continuum of hybrid male sterility is extremely rare (or unprecedented), as we said, and we think is quite surprising based on general understanding of the rate at which these forms of reproductive isolation usually evolve. This was our intended point. The reviewer correctly points out that the pattern of isolation is surprising given the low overall levels of genetic divergence. We have the modification to the manuscript to clarify this important point by emphasizing the pattern of reproductive isolation is noteworthy given genetic divergence (lines 210-220):

“Combining these broad population genomic patterns with our extensive crossing data provides two key insights into speciation. First, hybrid incompatibilities are accumulating extremely rapidly in this system. Intrinsic reproductive isolation tends to follow predictable patterns that are recapitulated in our crosses: incompatibilities manifest in the heterogametic sex first¹ sterility evolves more rapidly than inviability² and asymmetry precedes reciprocal isolation³. These evolutionary transitions have mostly been established through comparative analyses of crosses between different species pairs spanning a broad range of genetic divergences². The existence of population-level variation in hybrid incompatibilities is now well established³³; however, the striking progression in the strength and pattern of intrinsic reproductive isolation occurring between populations separated by such low levels of genomic divergence is highly unusual.”

Specific comments:

Line 68-78:

It is not clear how populations of mountain pine beetles are karyotypically different from each other. Line 72-73 describes the karyotype evolution between species (mountain pine beetle vs. Jeffrey pine beetle), this tells nothing about within species polymorphisms in karyotype.

Response: *There is no known within species karyotype polymorphism in the mountain pine beetle. We have added text to make that clear (lines 88-90)*

Line 73: “Despite low levels of genetic differentiation...”

I would like to see the actual numbers without searching for the ref 23.

Response: *The levels of genetic divergence we find are consistent with previous estimates and they are presented in our Results. We chose not to change the text here since this is a central Result from our paper.*

Line 86: It is not clear how the central boundary is defined.

Response: *We have added and modified the text to make this clear. (Lines 114-120).*

Line 90: It is unusual to consider “delayed development” as a character of hybrid inviability. In a strict sense, no matter how development is delayed, hybrids can be perfectly viable and fertile.

Response: *This is an important distinction. Delayed development almost certainly reflects intrinsic developmental issues in hybrid males, but the ultimate fitness effects are indeed extrinsic. Delayed development of hybrids has previously been described as extrinsic hybrid inviability by others (e.g., Rogers and Bernatchez 2006), and interpreting hybrid sterility and inviability phenotypes in the context of population ecology is common (Coyne and Orr 2004 spend a bit of time on this). We change the wording to make these distinctions clear (see lines 121-129)*

“We also detected asymmetric delayed hybrid male development in the most geographically distant cross (female AZ × male sCA; Fig. 2a and Supplementary Fig. 4). This transgressive hybrid phenotype likely reflects intrinsic developmental incompatibilities (i.e., HMI), but the fitness consequences depend on population ecology (i.e., extrinsic). Mountain pine beetles are univoltine (one generation per year) and the timing of their development is likely under strong selection in nature²⁶. Following emergence, beetles colonize heavily defended host pine trees by staging highly coordinated attacks and development time and emergence synchrony are crucial for beetle success. Thus, delayed development time would be highly maladaptive, thereby providing an ecologically relevant measure of hybrid viability²⁷.”

Line 91: “geographically distant cross”

I guess that an assumption for this pair of populations to be the most distant is that scattered small patches of host trees shown in grey in Fig 1a do not provide suitable habitat for the study species. In addition, the Great Basin Desert provides a geographic barrier. This should be described at somewhere in the text.

Response: *We have changed the text (lines 79-83 and figure caption for Fig. 2a) to make it clear that geographic and autosomal genetic divergence are correlated in the mountain pine beetle (Mock et al. 2007) and that the Great Basin Desert has likely been an important geographic barrier in this system.*

Another point is that why geographic distance was used and not the genetic distance. I think that the underlying assumption is ‘isolation-by-distance’, where the most geographically distant population pairs are also genetically distant. If so, it should be presented in more direct way. This is particularly important in light of ‘speciation-with-gene-flow’ because positive correlation between the level of genome-wide divergence and the strength of reproductive isolation is expected.

Response: *We have tried to make the interpretation of our crossing results in the context of genetic divergence more clear. Our PCA results (Fig 2c) for the autosomes show that genetic variation essentially loads on longitude (PC1) and latitude (PC2) and our population genetic metrics all show that sCA and AZ are the most genetically differentiated (Fig 2d and Supplementary Table 2). We have added text so that it is clear that genetic divergence ~ geographic distance when considering that gene flow occurs in a horseshoe-like pattern around the Great Basin Desert (see above).*

Finally, results of HMI for less distant population pairs should also be shown in Suppl.Fig3. For the same reason explained above, it is important to show weak or no HMI for less “genetically” distant populations.

Response: *This raises a good point that we have tried to clarify. For simplicity, we had summarized our overall interpretation of patterns of reproductive isolation based on our extensive crossing data (which has all been collected in a consistent manner), as well as other published data from this system. We now present all of the available crosses in Supplemental Figure 1, from which the detailed pattern of hybrid male sterility emerges. Unfortunately, development time data for both sexes is not available for all of our crosses and we know much less about this phenotype. This is because this extreme phenotype had never been detected and so was initially not part of our experimental focus. Development for both sexes has been recorded for some less geographically/genetically divergent crosses that did show symmetric HMS (i.e.,*

sCA and ID) and there was no evidence of delayed development in males (Bentz et al. 2011). Additionally, although not directly quantified, delayed development of hybrid males was not found in any of the Bracewell et al. 2011 crosses. The experimental methods used in Bracewell et al. (2011), entailed collecting emerging hybrid adults regularly and separating individuals by sex for subsequent backcrosses. Gross deviations in development time (like what was seen in the sCA x AZ cross) would have been noted and even slight deviations would likely have been apparent given the experimental design. We have added Supplementary Figure 1 to help make this clear.

Line 92: I do not understand this argument. Development timing for bivoltine insects should also be under strong selection.

Response: *We agree that development time is likely under selection in many/most insects. However, we are arguing that it is particularly intense in the mountain pine beetle given that reproductive success only occurs when exceptionally large numbers of beetles emerge synchronously and coordinate their attack on highly defended host trees. The importance of emergence synchrony in the mountain pine beetle has been the focus of a great deal of research and we have added text and citations to make this clear (lines 124-129).*

More importantly, it is very confusing to introduce development timing as a mechanism of reproductive isolation here, especially right after the result of HMI (delayed development of male hybrids). Are Line 91 and 92 in the same paragraph? It is unusual to consider temporal isolation as a form of hybrid inviability. Without showing results of HMI for OR and ID crosses, it seems pointless to discuss ecological segregation (or similarity in this case) between OR and ID.

I would suggest to present these results in a different way. First, there is little evidence for ecological segregation between OR and ID populations (and perhaps AZ and sCA, the pair shown in Suppl. Fig.3). However, AZ-sCA F1 hybrids show delayed development, which is a transgressive character (much more delayed than both parents). This could represent a strong barrier to gene flow. (Nonetheless, AZ and sCA populations appear to be geographically isolated for the first place, so it is not clear whether this temporal isolation plays any role in speciation).

Response: *We see now why this was confusing. As described above, we now clarify the interpretation of delayed development in the context of inviability. Following the reviewers suggestions, we have also revised the text as follows. First, we explain the population context of delayed development (Lines 132-135):*

“These populations (AZ and sCA) do not come into contact, thus delayed development may not currently contribute to reproductive isolation in nature but it does reveal another important component of evolutionary divergence leading to intrinsic incompatibilities between beetle populations.”

We then start a new paragraph interpreting the broad range of intrinsic incompatibilities (not just delayed development) in the context of expectations from ecological or phenotypic divergence (Lines 136-152). The main point is that these populations are isolated by hybrid male sterility and delayed development and these characters are cryptic with respect to ecological or phenotypic divergence.

Line 142-156:

Here the stringent filtering for sex chromosomes becomes problematic. Higher proportion of fixed sites on Y chromosome, for example, could be due to its smaller effective population size, stronger selection, as well as artifacts by removing ‘heterozygous’ sites. It is important to show how divergent X and Y gametologous regions. If they are young enough, then the vast majority of neoX and neoY is the pseudo-autosomal region (PAR), which behaves just like autosomes. Therefore, removal of het sites can induce serious bias toward fixed sites.

Response: *As outlined above, we understand these concerns thank the reviewer for pointing out the need to present more details on our SNP filtering. The decision to remove individual heterozygous sites and those with female coverage was based on first principles of the sex chromosomes (heterozygous sites do not exist on the male-specific portion of the Y), and we think that this is necessary and ultimately the correct treatment of these data. But we are also aware of the potential biases that such filters could introduce. We also now realize that adding the extent of neo-XY divergence is crucial to interpreting these data relative to other systems. In this revised manuscript we:*

- *Clarify our approach in the methods (lines 576-579, 587-589, 594-596, 619-622, 628-630, 638-650, 662-671, 710-716).*
- *Demonstrate that the general patterns of differentiation hold when we relax or remove X and Y filters. Regardless of treatment, both the rank order ($F_{st} \text{ neoY} \gg \text{ neoX} > \text{ ancX} \gg \text{ auto}$) and the extreme differentiation of the neoY hold (lines 710-716).*
- *Clearly explain that the identification of neo-X and neoY based on male:female coverage would exclude residual PAR regions (lines 171-173), which by definition appear as autosomes. Examination of retained PAR regions that still recombine between males and females, if they actually exist (see lines 91-95), would be interesting but this is a distinct partition from the X and Y specific regions of the sex chromosomes. The PAR is usually treated separately in papers focused on sex-linked reproductive isolation and usually shows much lower differentiation relative to the X and Y specific regions.*
- *We also more clearly define how divergent the neo-X and Y are in terms of sequence divergence (see below and lines 175-180).*

In addition, mtDNA is essentially one marker no matter how many SNPs it has. Thus, non-concordance with the ‘boundary’ could be a reflection of stochasticity.

Response: *We do realize that this is a single marker. The point here is not whether stochastic lineage sorting of the mtDNA affects phylogeographic patterns of variation, it almost certainly does. Rather, we are testing the hypothesis of if mtDNA variation can plausibly explain patterns of HMS. This is an a priori hypothesis that could explain the distinct asymmetric pattern of isolation we observed in our crosses. We can reject this hypothesis for mtDNA given the patterns of divergence in mtDNA haplotypes and the sharing of nearly identical mtDNA genomes between populations showing asymmetric reproductive isolation. We have modified the text to make the underlying hypothesis-testing framework clearer (line 154-159).*

One can infer the location of PAR (or less differentiated regions) by using read coverage

information.

Response: *As discussed above, it is unclear to us how we could use coverage information to identify PAR scaffolds in the fragmented MPB genome. True PAR scaffolds should have equal male and female coverage and would thus appear as autosomes. Most of the regions of neo-X and neo-Y scaffolds that have similar coverage between males and females (intervals with similar male:female coverage embedded in a male or female biased scaffold) appear to represent areas of low complexity. If we had a single build of each chromosome (a rare thing for most genome builds, and especially for neo-sex systems) then we might be able to physically localize apparent PAR regions. We are not there yet with this system.*

Line 208-209:

The difference between anc-X and neo-X may reflect the mappability of X-reads and Y-reads onto the X and Y reference genomes. Since there is no gametogs for anc-X, this region is less sensitive to the filtering process. However, neo-X is heavily influenced, resulting in the higher differentiation.

Response: *As mentioned above, we address these potential biases in a few ways. First, we show that the patterns are robust to sex-specific filtering. For example, F_{st} between the two OR and five ID females on the anc-X and neo-X without any male specific filters is 0.10 and 0.30, respectively (RAD-seq data). These estimates are slightly lower than the male estimates (with male-specific filters; anc-X = 0.16 and neo-X = 0.51) but the relative difference is nearly identical and the male and female results are qualitatively consistent and show that the neo-X is more differentiated than the autosomes and anc-X. Further, we explored the male only data and found that without applying any additional filters (thus allowing sites with female coverage and/or male heterozygous calls), that F_{st} for the anc-X, neo-X and neo-Y was 0.14, 0.30 and 0.48, respectively. Therefore, the relative differences between the genomic partitions are still readily apparent. Finally, we found that mapping qualities were similarly high for our filtered SNPs and intervals across partitions, although unfiltered mapping qualities were noticeably lower for the neo-sex chromosomes (now reported in Table 1) reflecting the repetitiveness of the neo-Y and potential cross mapping issues. We have added text and results to the methods to make clear the difficulties of filtering SNPs on neo-sex chromosomes and degenerate neo-Y's (See lines above).*

Line 253-254: “These patterns are seemingly at odds with our inferences of sex-linked HMS and reduced gene flow.”

Lower D_{xy} for X and Y is not surprising given the smaller N_e for sex linked markers. ($D_{xy} = 2ut + 4N_eu$ for diploid markers. See Nachman and Payseur 2012)

Response: *Good point, with was poor wording on our part. We were anticipating that readers might be surprised by the lower sex-linked divergence, but we may have belabored the point. We have removed this statement and shorten the section on divergence (see lines 262-296).*

Line 309: “The neo-Y was highly enriched for male-specific genes (Fig. 4a and Supplementary Fig. 10).”

Supplementary Fig. 10 does not have plot for neo-Y.

Response: *We have fixed the error.*

Line 446: two reference genomes with 3 accessions?

Response: *We changed the text to make clear that these are not reference genomes accessions. These are the deposited raw male and female reads that were used to assemble the reference genomes by Keeling et al. 2013 and that we reanalyzed to identify sex-linked scaffolds using coverage based methods (Lines 532-535).*

Line 457-458:

Clarify how male-to-female ratio is calculated. For X chromosome female/male results in $>2x$, but not for Y if one takes female/male (and hence must be male/female??)

Response: *We used an approach that quantified read coverage using EdgeR and framework similar to how fold changes in gene expression are calculated, leveraging the fact that both are essentially count data. This is a natural extension of the idea of male to female coverage ratio but a bit more sophisticated and defines the sex-differences as differences in log fold changes in sequence coverage. (line 540-548)*

Line 487: “this bias”

Which bias?

Response: *We changed the text for clarity (line 574).*

Line 490: “confident genotype calls”

Please describe. Are these synonymous to the sites that passed the filter?

Response: *Yes, these are synonymous with the sites that passed the filter and we included text to make this clear (line 581). For STRUCTURE, PCA, Fst and ABBA-BABA tests we only used sites where we could accurately genotype 48% or more individuals at the site. This was driven by the fact that below these thresholds, there was an increased chance that all individuals within a population ($n = 3$ per population) were not genotyped.*

Line 491-496:

I am seriously concerned about potential artifacts introduced by the SNP filtering on X and Y chromosomes. It is not clear when the neo-XY chromosomes are formed and how divergent they are, but I assume that they have relatively recent origin with substantial sequence similarity. The fate of newly formed neo-XY chromosomes is described elsewhere (see Bachtrog papers). Briefly, at the onset of X-autosome fusion, the whole neo-X and neo-Y are essentially the pseudo-autosomal region (PAR). As recombination is suppressed, the size of PAR reduces and differentiation increases between neo-X and neo-Y. The pine beetle system represents some intermediate stage of sex chromosome differentiation with a number of gametologous reads being mapped to the alternate sex chromosomes. Therefore, these filter sets can remove less divergent regions including PAR and choose highly differentiated regions only.

Response: *There is currently no estimate of when the neo-sex chromosomes formed. However, both the mountain pine beetle and its closest relative, the Jeffrey pine beetle, appear to share the fusion, which does give us some estimate of a minimum age (Lines 182-183). To better address how divergent the neo-sex chromosomes are, we have added text and a table which highlights how fragmented the Y chromosome appears in the genome build in contrast to other genomic partitions (Table 1). We have also calculated Ks for 119 neo-X/neo-Y homologous genes. Ks estimates suggest that the neo-X and neo-Y are moderately divergent (mean Ks = 5.7%, median = 2.2%) and together with the assembly issues for the neo-Y suggest that the neo-sex chromosomes are of moderate age. We have added text to the manuscript (lines 170-183).*

Line 501: 29? Not 27?

Response: *Correct, the PCA was with 29 (as were other analyses). This included the 27 we resequenced plus the 2 CAN reference individuals.*

Line 523:

If the same coverage thresholds were used for Y with autosomes, it is in practice coverage >6x-equivalent in autosomes.

Response:

The purpose of the $\geq 3x$ and ≤ 1 SD coverage filter was to select subsets of the genome where we could feel fairly confident about making both variant and invariant genotype calls across all genomic partitions (anc-X, neo-X, neo-Y and autosome) given that coverage would differ because of ploidy. We explored what influence different lower thresholds might have on our results (i.e., double the lower coverage threshold for autosomes) and qualitative patterns remained consistent. So for simplicity, we maintained the 3x threshold across all partitions.

Line 526:

Male het sites on Y-chromosome also represent Y-regions that are not diverged from X-gametologous regions. Similar to the stringent filter that the authors used, this filter removes, true Y regions with lower divergence. Thus the bias introduced by their stringent filter (Line 519-520) cannot be removed.

Response: *We think we have addressed most of this above and we have rewritten the methods to be clearer. The general patterns of between population differentiation hold even when the filters are removed completely. We have shown that the neo-X and Y are perhaps more genetically divergent than is seen in some neo-XY systems such as *D. miranda*. We note that the reviewer is correct in that we are filtering the data in a number of ways to identify regions of the neo-X and neo-Y that are sufficiently differentiated that we can analyze them. Again, key patterns hold when we remove these filters, but the goal here is indeed to identify X and Y-linked regions that are free from cross mapping issues. We are not focused on (or particularly interested in) the PAR, if it exists. This seems reasonable to us and consistent with most studies of sex-linked isolation which often exclude the PAR. Our analyses suggest that most of this noise reflects low complexity (collapsed regions in the build, especially on the Y) but there are certainly some cross mapping of reads between the X and Y. Male heterozygous regions and female coverage are both reasonable metrics to do this.*

For the neo-Y, it is unclear to us how our filters would strongly bias us towards SNPs that are variable within mountain pine beetles, but for which all variants occur between populations (i.e., $F_{st} = 1$). One could worry that our filters for regions that appear clearly differentiated between the X and Y would bias us towards regions of the X and Y that have the highest substitution (mutation) rates, and hence introduce a bias towards higher divergence estimates. We note that the neo-X and neo-Y both show relatively low levels of variation and divergence, which is opposite of what might be expected if we are somehow strongly selecting for a subset of very rapidly evolving sex-linked regions.

Line 543:

There are a number of publicly available tools to detect indels and other structural variations with much more sophisticated approaches by combining read-map, split-read, and read-depth information (e.g., Manta SV, Lumby, Delly, Breakdancer, etc. see Pirooznia et al. 2015 for comprehensive review). I would recommend to use some of these tools to quantify SV frequency and density across the genome and compare it between auto-x-y chromosomes.

Response: *We appreciate the recommendation and we have now attempted to use two different methods (Lumpy and Pindel) to help quantify the number of SV's in the different genomic partitions. Unfortunately, these tools seem better suited for less extreme structural variation and for more contiguous genome builds. We managed to get estimates of indel variation using Pindel that were qualitatively in line with our results from our coverage-based analyses and suggest that the neo-Y shows significant structural differences between east and west beetles. We have decided not to present these results because in spot-checking the data we found numerous inconsistencies. A serious difficulty in doing this is that the published reference genome, and particularly the scaffolds we identified as neo-Y chromosome, is comprised of many small contigs (see Table 1) that are heavily scaffolded together (see Keeling et al. 2013). This fragmented structure, combined with assembly errors and enrichment for repetitive sequence, seems to have a negative affect on the performance of these more sophisticated tools. That is why we had focused our initial analyses on gene deletions using simple sequencing coverage. Although our approach based on differential read coverage is simple, we believe it is a conservative method with respect to these assembly issues.*

Line 544:

What is 'high quality scaffolds'?

Response: *We deleted 'high quality' as this should be clear that these are the same scaffolds as described in earlier sections (line 674).*

Line 547: "pheatmap"

Typo?

Response: *No, the R package name is indeed pheatmap. It is short for Pretty Heatmap.*

Figure 1C, Admixture

No Axis labels?

Response: Fixed, see Reviewer 1 comments.

Supplementary Table 2:
This is a figure and not table.

Response: We have now converted the figure into a table (see Supplementary Table 2).

Supplementary Figure 2A: Use larger font size. X-axis labels, in particular, are not legible. The scale in Y-axes is not described. Are these the same with the one in 2B? I guess not. Colour code (red, yellow, green) is not described.

Response: Thanks for the suggestion. We made fonts larger and added text to the figure caption for clarity and directed the reader to the Methods where we describe how we determined sperm quantity for the different structures (See supplementary Fig 3)

Supplementary Figure 3:
“The dashed [vertical] line represents the median development time for each sex.”. They should be colour-coded for different sexes.

Response: Thanks for the suggestion. We have added color to the figure.

Supplementary Figure 5:
Methods describes log-fold change as female/male ratio (Line 456-459), while this figure appears to show male/female. Use one of the other consistently.

Response: Partially addressed above. We have added text to the figure caption to make this more clear (Supplementary Figure 6)

Reviewers' comments:

Reviewer #2 (Remarks to the Author):

I reviewed this MS previously and my main comment addressed whether this very nice population genomics study provides evidence of a link between neo-sex chromosome formation and speciation. The study provides great data on neo-sex chromosome degeneration and great data on reproductive isolation between some populations (according to a geographical pattern that was already known from previous published studies). However, I still believe that it is not possible to conclude that the neo-sex chromosome drives speciation. Indeed, the authors have weakened their statements somewhat in the revision but the take-home message in title, abstract and conclusions is a link between neo-sex chromosome and speciation.

OR*ID shows only weak asymmetric HMS (>80% are not showing HMS according to the supplements), but neo-Y is more or less completely fixed (F_{st} close to 1; Figure 3). This suggests that neo-Y is not driving HMS (contrary to conclusions). Although the neo-Y is differentiated and still express RNA on some genes it does not at all need to be involved in HMS. If the neo-sex chromosome is old it can certainly be so that neo-Y has very little adaptive significance (RNA is produced but no functional proteins) and that neo-X and anc-X play a much more prominent role. The X-chromosome recombines and is expected to have less deleterious mutations than Y. Also, the dominance theory of Haldane's rule rather suggests that recessive X-linked genes are causing HMS (recessive deleterious mutations are being exposed in the hemizygous sex (XY, males) and/or the recessive functional X-alleles that are inherited from the mother interact with dominant mutations inherited from the father elsewhere in the genome). There are probably several X-linked genes that are highly differentiated and may be involved in HMS/HMI. The results show that the average F_{st} on anc-X and neo-X is lower than for neo-Y but still very high, and high enough to account for <20% HMS. This means that one cannot conclude whether neo-Y, neo-X or anc-X is driving HMS/HMI.

Other comments:

Only some crosses are illustrated in Figure 2a (compare Figure 2a and Suppl Fig 1). Figure 2a needs to show all crosses and the outcome in terms of reproductive isolation (e.g. with dashed lines for population pairs with no reproductive isolation). You should make it clear which crosses are new for this study and which are not.

Moreover, judging from Figure 2d, also other population pairs, including some populations in east, will show more or less completely fixed neo-Y; thus, detailed population genomics studies also between other populations are recommended.

In the text, you refer to geographical distance, but mean distance over the HMS-boundary.

Question: When calculating $\log(\text{obs}/\text{exp})$ in Figure 4, were you accounting for the fact that anc-X, neo-X and neo-Y are hemizygous in males? And what about gametologous genes on

neo-XY that are similar so that reads might map to both neo-X and neo-Y?

Reviewer #3 (Remarks to the Author):

This is my second revision of this manuscript. I really appreciate the effort that the authors made by addressing questions and concerns that I raised in the first revision. The current version of ms presents much more balanced view with respect to the causal relationship between sex chromosome evolution and speciation and lays a solid foundation for further studies to investigate genetic basis of reproductive isolation by focusing on neo-sex chromosomes. Presenting lower bound of neo-sex chromosome age is an important addition coupled with possible 'distance-pairing heteromorphic sex chromosomes', which I completely missed in the previous ms. This additional information cleared up many concerns in genotyping.

I do not have much further comments to offer (see below). I look forward to the follow-up studies in this unique system in speciation genomics.

Specific comments:

Line 61-68

I like the addition of new figure 1. However I am not fully convinced that the sequence of events described here depicts the general pattern of evolution of reproductive isolation. Specifically, I am wondering if emergence/establishment of reproductive isolation by auto-sex chromosome fusions is a general (or inevitable) fate and essential first step. Is it fair to assume that auto-sex chromosome fusions result in reproductive isolation, and the evolution of this reproductive isolation mechanism precedes changes associated with recombination cessation (ie., gene degeneration, sex-specific functionalization, etc.)? Examples shown here (ref 16 and 17) might be a special case where linkage of genes associated with reproductive isolation (eg., mate preference) with sex-limited chromosomes (Y or W) plays an important role, and fusions per se may not drive speciation. Second, 'changes in chromosome structure, gene content, and expression underlying the evolution of sex-specific functions' could also lead to the emergence of reproductive isolation, as the authors discussed later in the ms. This is just a matter of how 'three general stages' are phrased. The current description would give a false impression about the timing of the evolution of reproductive isolation mechanisms in association with neo-sex chromosome evolution.

Line 64:

Sticklebacks and 'butterflies'.

Line 91:

The term 'distance-pairing heteromorphic sex chromosomes' was added in this version of ms. I think that this is a good idea, but a bit more explanation might be needed because

this is not a widely used term. Why do not they synapse? How do X and Y chromosomes achieve proper segregation without synapse formation? On a related note, 'Xyp' notation in Figure 2b also requires some explanations in the figure legend (why 'y' is a lower case, and what 'p' stands for: ref 12 says 'In the Coleoptera cytogenetic literature, distance-pairing sex bivalents are usually denoted with a lower case letter that describes how they are oriented during meiosis (e.g., "p" stands for parachute and indicates a large X chromosome with a small Y chromosome that appears suspended from it').

Line 156-159 'This distinct genetic architecture requires epistatic interactions involving genetic factors with uniparental inheritance³. In mountain pine beetles this equates to nuclear-cytoplasmic interactions involving mitochondrial DNA (mtDNA) and/or sex chromosome-linked interactions.'

Is it also worth mentioning possible involvement of cytoplasmic endosymbionts (eg., Wolbachia) in asymmetric RI? It is partly consistent with the asymmetric fertility reduction in ORxID crosses and nCAxMT crosses if one assumes endosymbiont infection in eastern populations. This does not explain bidirectional RI in more distant crosses, nonetheless.

Line 174: '...more repetitive...'

Are there any statistics shown in Table 1 and Supp Fig 6? It seems that the authors have annotated repetitive sequences in the genome (eg., '...a large portion of filtered SNPs occurred in repetitive regions or stretches of low sequence complexity [line 589]'). If repeats were annotated (by RepeatModeller, etc), please describe it accordingly both in methods and results.

Additional information that the authors might want to present is repeat (transposon) density in different parts of the genome if repetitive sequences were annotated. The authors described the existence of putative loss of function mutations on neo-Y by indels (line 392). Since accumulation of transposable elements on non-recombining Y is an inevitable fate, showing quantitative measurement of repeat contents would further strengthen the conclusion.

Reviewers' comments:

Reviewer #2 (Remarks to the Author):

I reviewed this MS previously and my main comment addressed whether this very nice population genomics study provides evidence of a link between neo-sex chromosome formation and speciation. The study provides great data on neo-sex chromosome degeneration and great data on reproductive isolation between some populations (according to a geographical pattern that was already known from previous published studies). However, I still believe that it is not possible to conclude that the neo-sex chromosome drives speciation. Indeed, the authors have weakened their statements somewhat in the revision but the take-home message in title, abstract and conclusions is a link between neo-sex chromosome and speciation.

Response: We thank the reviewer for their enthusiastic comments regarding aspects of our study. We emphasize the three key general insights provided by our study, which we believe represent sufficiently important conceptual and empirical advances in sex chromosome evolution and speciation that rise to the high standards of Nature Communications.

1) Our manuscript demonstrates an unprecedented geographic continuum of rapidly evolving

intrinsic reproductive isolation. These striking patterns are clearly linked to the sex chromosomes, have evolved independent of apparent ecological drivers, and are outpacing other typically rapidly evolving isolating mechanisms (e.g., our novel characterization of pheromone divergence in this system - Supplementary Fig. 5). We maintain that these results are highly novel, regardless of how they specifically connect to neo-sex chromosome evolution. Further, though some of the crossing data has been previously published, we strongly disagree with the conclusion that the full geographic pattern was already known. At best, Bracewell et al. (2011) was able to infer that there was some geographic variation in hybrid male fertility. This had not been directly linked to a breakdown of spermatogenesis (Supplemental Figure 3), nor had the complete pattern of a continuum of intrinsic reproductive isolation between several population pairs been described. It was also not known that the most distant cross manifests hybrid sterility and inviability (Supplementary Figure 4). Finally, the extremely close genetic relatedness among these populations was not known, precluding insights into the degree of divergence underlying these patterns.

2) We provide several important insights into the tempo of neo-sex chromosome evolution, including the relative rates of sex-limited gene specialization and structural degeneration between the neo-X and neo-Y chromosomes. Very few studies have examined the dynamics of specialization and degeneration in neo-sex chromosome systems and none have attempted to link these hallmarks of sex chromosome evolution to population-level variation through extensive genomic sequencing and genetic crossing data.

3) We feel strongly that our presentation establishes a convincing connection between rapid neo-sex chromosome evolution and functional divergence between populations, and that these patterns have clear implications for speciation. We thoroughly outlined the evidence supporting our inferences in the response letter of the previous revision and will not revisit all of these points here. We have unequivocally demonstrated that reproductive isolation is rapidly evolving and sex-linked. We maintain that our extensive genetic crossing data, population genomic comparisons, and functional genomic characterization collectively provide consistent and strong support an involvement of the neo-sex chromosomes in reproductive isolation, with particularly strong support for an involvement of the neo-Y (no gene flow/admixture, extreme structural divergence, and considerable divergence in spermatogenic gene contents between populations isolated by hybrid male sterility).

We certainly did not mean to imply that there could not be an important contribution of the X to reproductive isolation. But we do feel that it would be very tenuous to argue that all of the patterns that we describe could plausibly be explained through the ancestral X chromosome given signatures of much lower genetic differentiation (more than 3-fold lower compared to the neo-X, 6-fold lower when compared to the neo-Y) and considerable admixture on this chromosome. Indeed, our expanded population-level sampling suggests that reproductive isolation is more strongly linked to the neo-X portion of the X chromosome (see Ln 321-332). Although we appreciate that aspects of our inferences are indirect, and we acknowledge that genotype-to-phenotype connections are unresolved, we also are at a loss to come up with plausible alternative interpretations of these data that do not include a central role of the neo-sex chromosomes.

Our central arguments remain as before, but we have tried to address the reviewers concerns in a number of ways:

- We have added a more complete discussion of the genetic architecture of reproductive isolation, including a more balanced treatment of the possible contributions of the neo and ancestral X chromosomes throughout and the contributions of related X-linked dynamics (e.g., dominance theory). See LN 102-105; 275-278; 327-332; 429-438; 452-456; L457-467*

- We have added text discussing the breakdown of reproductive barriers in the face of gene flow, and how this might relate to the evolution of the sex chromosomes. L341-357; L454-456.
- We have added additional text on the biogeography of allopatry and historical refugia. L333-340.

OR*ID shows only weak asymmetric HMS (>80% are not showing HMS according to the supplements), but neo-Y is more or less completely fixed (F_{st} close to 1; Figure 3). This suggests that neo-Y is not driving HMS (contrary to conclusions).

Response: We believe that we were unclear in presenting this phenotype, leading to the incorrect conclusion that sterility in this population is highly polymorphic and thus at odds with the conclusion that the neo-Y is involved in reproductive isolation. Supplemental figure 2 showed the proportion of males that were completely sterile (~20%). Here 'fertile' males are defined according to a very conservative criterion defined as those that fertilized 1 or more eggs in test crosses. This does not imply that males with some reproductive output are fully fertile. Rather these males appear to have substantially reduced fertility and overall hybrid male fecundity appears significantly reduced in this population across phenotyped males (to the limits of this fertility assay). This pattern is not at all at odds with Y-linked incompatibilities. Some variation in the degree of male sterility is a hallmark of HMS. Sterility is a notoriously complex phenotype and is undoubtedly sensitive to male condition, environmental effects, etc. Thus, our results do not provide evidence for polymorphism in HMS in this cross direction. We have clarified this with edits to the text (Ln 123-127) and to Supplementary Figure 2, where we clarify this conservative criterion of sterility and now also present additional data on average male fecundity (total eggs hatched in test crosses) for all males in reciprocal crosses. These added data clearly show the strong asymmetric reduction in fertility of the hybrid males in question (female ID x male OR).

Although the neo-Y is differentiated and still express RNA on some genes it does not at all need to be involved in HMS. If the neo-sex chromosome is old it can certainly be so that neo-Y has very little adaptive significance (RNA is produced but no functional proteins) and that neo-X and anc-X play a much more prominent role.

Response: The neo-sex chromosomes have existed for at least a few million years (Ln 193-94) and show some genetic differentiation relative to the neo-X (median $K_s = 2.2\%$), but it is misleading to call this an old sex chromosome system. Though there is clearly extensive gene specialization and structural variation, nothing in our genomic or transcriptomic data suggest that the neo-Y can be dismissed as of little adaptive (or functional) significance. Based on current annotation there are 781 genes predicted on the neo-Y (Table 1), and we have verified expression of 100s of these. Many of these genes are very highly expressed in males (testis and or heads). Though RNA expression does not assure protein translation, it seems highly unlikely that hundreds of neo-Y genes with intact coding frames show high tissue specific transcription with no translation to functional proteins.

The X-chromosome recombines and is expected to have less deleterious mutations than Y. Also, the dominance theory of Haldane's rule rather suggests that recessive X-linked genes are causing HMS (recessive deleterious mutations are being exposed in the hemizygous sex (XY, males) and/or the recessive functional X-alleles that are inherited from the mother interact with dominant mutations inherited from the father elsewhere in the genome). There are probably several X-linked genes that are highly differentiated and may be involved in HMS/HMI. The results show that the average F_{st} on anc-X and neo-X is lower than for neo-Y but still very high, and high enough to account for <20% HMS. This means that one cannot conclude whether neo-Y, neo-X or anc-X is driving HMS/HMI.

Response: As outlined above, the most striking empirical links to reproductive isolation derive from the neo-Y. We do not exclude contributions of the X chromosome. Indeed, we expect that X-Y interactions are likely crucial to the evolution of reproductive isolation in this and other neo-sex chromosome systems. Our inference that the Y chromosome is involved in reproductive isolation in these crosses is not at odds with the dominance theory of Haldane's rule. Indeed, all available theory suggests that in XY systems the Y may indeed be an important contributor to Haldane's rule. Like the X, the Y chromosome is hemizygous in males and thus shows immediate exposure of recessive incompatibilities that interact elsewhere in the genome (the X or the autosomes). Moreover, there is considerable evidence for a large-Y effect in speciation (see below). We now include a bit more discussion of the nature of such interactions (see Ln L341-350; 426-438) and some more general reference to dominance theory in the context of neo-XY systems (see Fig. 1 legend).

The primary emphasis on the X chromosome in Turelli and Orr's classical mathematical treatment of dominance theory (Genetics, 1995) was done in part for modeling convenience, and later formally expanded upon to include the Y (2000). The strong bias towards emphasizing the X in much of the early foundational literature reflects the extremely small gene content on the Drosophila Y and the fact that the Y is less likely to be involved in inviability in this system because it is primarily expressed in the male germ line. On first principles of dominance theory, the Y is generally predicted to play an important role in Haldane's rule and the evolution of HMS. Here are some statements from Turelli and Orr's work in support of this interpretation:

Summarized from Turelli and Orr (1995), p397

"Although the Y chromosome has no essential somatic function in Drosophila (Ashburner 1989), it is required for fertility in most species.....Although we have focused on X-autosomal interactions, there are clearly many more ways of evolving hybrid male sterility than inviability. Hybrid sterility can result from X-autosomal, autosomal-autosomal, cytoplasmic-autosomal, X-Y, Y-autosomal, and Y cytoplasmic incompatibilities..... Genetic analysis shows that the Y does, in fact, often play a major role in hybrid sterility in Drosophila (reviewed in Coyne and Orr 1989a; Johnson et al. 1993). No cases are known, however, where the Y affects hybrid viability in Drosophila."

Turelli and Orr (2000) also provide a fairly thorough treatment of the contribution of the Y to reproductive isolation and conclude that Y effects on postzygotic isolation are extraordinarily common (see pg 1670 and table 1 for empirical support and 1670-1671 for theoretical support). From these models and available empirical data the effect of the Y on Haldane's rule should scale with size and gene content. There is no formal treatment in neo-sex systems. However, these simple relationships suggest that neo-Y chromosomes, which are comparably large and contain more genes than old heteromorphic sex chromosomes, are likely to be important in the evolution of HMS.

Other comments:

Only some crosses are illustrated in Figure 2a (compare Figure 2a and Suppl Fig 1). Figure 2a needs to show all crosses and the outcome in terms of reproductive isolation (e.g. with dashed lines for population pairs with no reproductive isolation). You should make it clear which crosses are new for this study and which are not.

Response: Inclusion in Figure 2a leads to an exceptionally messy presentation. These details are available in Supplemental Figure 1. However, we have edited the legend to directly reference these additional crosses so as to avoid confusion.

Moreover, judging from Figure 2d, also other population pairs, including some populations in east, will show more or less completely fixed neo-Y; thus, detailed population genomics studies also between other populations are recommended.

Response: We have pointed out the limitations of the current data with respect to the east and southeast populations (Ln 235-241). We agree that additional insights into the full pattern of reproductive isolation will be gained with additional genetic and population genomic studies focused on this part of the range.

In the text, you refer to geographical distance, but mean distance over the HMS-boundary.

Response: Yes, that is correct. We have edited the text to more precisely describe this context when we first introduce the pattern (Ln 121-131):

“..the severity and F_1 architecture of HMS consistently changed with geographic distance of hybridizing population pairs found on either side of a central East-West boundary of reproductive isolation. Crosses between proximate populations on either side of this boundary resulted in hybrid males with reduced fertility (Supplementary Fig. 2) in one direction of the cross (i.e., weak asymmetric HMS; female ID \times male OR), while reciprocal crosses between more distant localities resulted in severe or complete HMS (i.e., strong symmetric HMS; Fig. 2a). We also detected asymmetric delayed hybrid male development in crosses between beetles from Arizona (AZ) and Southern California (female AZ \times male sCA; Supplementary Fig. 4), which is the most geographically distant cross type based on the distribution of host trees (Fig. 2a).”

The context of a central boundary of reproductive isolation is reinforced in Figure 2 and throughout the main text. For brevity we refer to this more complex pattern as increased isolation with geographic distance elsewhere in the manuscript.

Question: When calculating $\log(\text{obs}/\text{exp})$ in Figure 4, were you accounting for the fact that anc-X, neo-X and neo-Y are hemizygous in males? And what about gametologous genes on neo-XY that are similar so that reads might map to both neo-X and neo-Y?

Response: These are enrichment tests of genes detected as expressed in a given tissue. The statistics are based on counts of genes. We have clarified this in the legend and in the methods (Ln 770-775, see also Supplemental Table 5). Results based on gene counts were consistent across minimum thresholds of > 1 and 10 FPKM and we did not modify these values across partitions. We have not thoroughly explored issues of dosage compensation in this system. Regarding gametologous genes, our results show (Ln 195) that many of the neo-X and neo-Y genes are sufficiently divergent and therefore most reads should map uniquely.

Reviewer #3 (Remarks to the Author):

This is my second revision of this manuscript. I really appreciate the effort that the authors made by addressing questions and concerns that I raised in the first revision. The current version of ms presents much more balanced view with respect to the causal relationship between sex chromosome evolution and speciation and lays a solid foundation for further studies to investigate genetic basis of reproductive isolation by focusing on neo-sex chromosomes. Presenting lower bound of neo-sex chromosome age is an

important addition coupled with possible ‘distance-pairing heteromorphic sex chromosomes’, which I completely missed in the previous ms. This additional information cleared up many concerns in genotyping.

I do not have much further comments to offer (see below). I look forward to the follow-up studies in this unique system in speciation genomics.

Specific comments:

Line 61-68

I like the addition of new figure 1. However I am not fully convinced that the sequence of events described here depicts the general pattern of evolution of reproductive isolation. Specifically, I am wondering if emergence/establishment of reproductive isolation by auto-sex chromosome fusions is a general (or inevitable) fate and essential first step. Is it fair to assume that auto-sex chromosome fusions result in reproductive isolation, and the evolution of this reproductive isolation mechanism precedes changes associated with recombination cessation (ie., gene degeneration, sex-specific functionalization, etc.)? Examples shown here (ref 16 and 17) might be a special case where linkage of genes associated with reproductive isolation (eg., mate preference) with sex-limited chromosomes (Y or W) plays an important role, and fusions per se may not drive speciation. Second, ‘changes in chromosome structure, gene content, and expression underlying the evolution of sex-specific functions’ could also lead to the emergence of reproductive isolation, as the authors discussed later in the ms. This is just a matter of how ‘three general stages’ are phrased. The current description would give a false impression about the timing of the evolution of reproductive isolation mechanisms in association with neo-sex chromosome evolution.

Response: We present this general model (Fig. 1) as testable hypotheses to help frame the problem but we see how aspects of our general presentation might have been misleading or overly general. Simple chromosomal isolation between a newly established sex chromosome system and the ancestral system is arguably the earliest possible mechanism of isolation. Also, it is not a given that the establishment of a neo-sex chromosome system will inevitably lead to isolation due to chromosomal incompatibilities in meiosis. We also did not intend to imply that isolation was chromosomal in sticklebacks and butterflies, just that these systems were in the early stages of neo-sex chromosome evolution. In this revised manuscript we have edited reference to these important previous results and clarified the uncertainty associated with these general stages as follows:

Ln 60-62 - First, the establishment of a new sex chromosome system within a population could lead to reproductive isolation between species with different sex chromosome systems.

Ln 68-71 - In principle, sex chromosome transitions could also result in rapid functional divergence and the accumulation of genetic incompatibilities between populations, but the tempo of these genomic changes and their contribution to broader patterns of species diversification remains largely unknown.

Ln 71-74 - For example, genes involved in reproductive isolation have been linked to recently established neo-sex chromosomes in threespine sticklebacks¹⁶ and butterflies¹⁷, but it is unclear if or how these hybrid incompatibilities relate to neo-sex chromosome evolutionary dynamics per se.

We have also clarified the contribution of dominance theory and faster-X evolution to reproductive isolation in neo-XY systems in the legend of Fig. 1 (see also response to Reviewer 2):

Ln 1062-1065 - As seen in highly heteromorphic sex chromosomes, the evolution heteromorphic neo-sex chromosomes is also predicted to further contribute to reproductive isolation through the exposure of recessive genetic incompatibilities and more rapid sex-linked divergence in the heterogametic sex.

Line 64:

Sticklebacks and ‘butterflies’.

Response: Fixed.

Line 91:

The term ‘distance-pairing heteromorphic sex chromosomes’ was added in this version of ms. I think that this is a good idea, but a bit more explanation might be needed because this is not a widely used term. Why do not they synapse? How do X and Y chromosomes achieve proper segregation without synapse formation? On a related note, ‘Xyp’ notation in Figure 2b also requires some explanations in the figure legend (why ‘y’ is a lower case, and what ‘p’ stands for: ref 12 says ‘In the Coleoptera cytogenetic literature, distance-pairing sex bivalents are usually denoted with a lower case letter that describes how they are oriented during meiosis (e.g., “p” stands for parachute and indicates a large X chromosome with a small Y chromosome that appears suspended from it’).

Response: Thanks for catching this omission. We have added clarification of the degenerate nature of the Y (Ln 93) and explained the parachute terminology in the legend of Figure 2 (Ln 1080-1082). Regarding synapsis and proper segregation, the basic molecular mechanisms underlying this common form of sex chromosome pairing remain surprisingly unknown.

Line 156-159 ‘This distinct genetic architecture requires epistatic interactions involving genetic factors with uniparental inheritance³. In mountain pine beetles this equates to nuclear-cytoplasmic interactions involving mitochondrial DNA (mtDNA) and/or sex chromosome-linked interactions.’

Is it also worth mentioning possible involvement of cytoplasmic endosymbionts (eg., *Wolbachia*) in asymmetric RI? It is partly consistent with the asymmetric fertility reduction in ORxID crosses and nCAxMT crosses if one assumes endosymbiont infection in eastern populations. This does not explain bidirectional RI in more distant crosses, nonetheless.

Response: This is a good point. In our previous study (Bracewell et al. 2011), we showed that patterns of F1 fertility in asymmetric crosses were consistent in reciprocal backcrosses. That is, F1 males were consistently fertile or sterile in test crosses to females from either source population, which would not be expected under a standard model of Wolbachia-induced incompatibilities. We have also failed to detect Wolbachia in reproductively isolated populations using standard PCR screens from whole beetles. Though such molecular diagnostics are not definitive, this combined with the backcross results and the bidirectional RI in more distant crosses really argues against this alternative explanation. We have edited the text as follows to clarify these issues:

Ln 165-167: Further, there is no evidence for variable endosymbiont infections (e.g., Wolbachia) that might contribute to asymmetric HMS in this system based on standard genetic and molecular diagnostics²⁴. Thus, this distinct genetic architecture requires....

Line 174: ‘...more repetitive...’

Are there any statistics shown in Table 1 and Supp Fig 6? It seems that the authors have annotated repetitive sequences in the genome (eg., ‘...a large portion of filtered SNPs occurred in repetitive regions or stretches of low sequence complexity [line 589]’). If repeats were annotated (by RepeatModeller, etc), please describe it accordingly both in methods and results.

Additional information that the authors might want to present is repeat (transposon) density in different parts of the genome if repetitive sequences were annotated. The authors described the existence of putative loss of function mutations on neo-Y by indels (line 392). Since accumulation of transposable elements on non-recombining Y is an inevitable fate, showing quantitative measurement of repeat contents would further strengthen the conclusion.

Response: These were intended to be qualitative statements regarding the overall patterns in Table 1 and Supplemental Figure 6. We have explored various aspects of repeat content derived from available genome annotations based on RepeatMasker and WindowMasker, but we have found these analyses to be noisy with respect to comparisons among partitions due to the rather fragmented build of the neo-Y. Many repeats appear to be collapsed (resulting in very high coverage) or at the end of contigs (resulting in more fragmentation and smaller contigs). Our preliminary analyses of repeat content seem to reflect the limitations of the build as much as the actual biological differences between the partitions. Most of our comparative inferences of repetitive content and complexity have been qualitative in nature based on coverage. Given the current genomic resources, we would rather not include a thorough treatment of repeat content in the current study. We have edited the text to remove the quantitative statement regarding repetitive content (Ln 185), and we have clarified the statements regarding filtered SNPs as follows (Ln 614-617):

“The neo-sex chromosomes were not repeat-masked in advance and a large portion of filtered SNPs occurred in repetitive regions or stretches of low sequence complexity based on variance in local sequencing coverage and lower mapping qualities”

REVIEWERS' COMMENTS:

Reviewer #2 (Remarks to the Author):

Finding the genetic basis for speciation is not a trivial task and somewhat of the Holy Grail in speciation research. It is impossible to get conclusive evidence regarding processes (i.e. speciation) from genomic patterns. In this article, the authors make too strong statements about causality. I strongly disagree that these results provide evidence that this neo-sex chromosome (which is much older than the speciation event) drives speciation. Also, they draw too strong emphasis on the neo-Y: the neo-X and to some extent ancestral-X are also highly differentiated (yes, I know, not as highly as neo-Y, but nevertheless a DM-incompatibility may act on just a few genes or SNPs, and you cannot dismiss all potential interactions involving neo-X or ancestral-X with your data). What you have found however is that strong differentiation of sex chromosomes coincides with reproductive isolation. To me this is an amazing finding that could be followed up in future research (e.g. with functional genomics of sex chromosome-linked genes; knockouts, etc.) to reveal the potential causality between sex chromosome-linked genes and speciation. I am impressed by the data and results in this paper, but unimpressed by the unnecessary overstatements in the title and abstract and elsewhere.

Title, should read:

Recent sex chromosome evolution *coincides* with speciation in a major forest pest

Abstract; the final sentence should be rephrased:

Our result *suggests* that neo-sex sex chromosome evolution *may* drive rapid...

(By the way, this neo-sex chromosome was formed long before this particular speciation event, but you state in the final sentence ...that early stages of neo-sex chromosome evolution can drive... You do not mean *early* I assume (or perhaps you mean early states of sex chromosome evolution rather than early stages of neo-sex chromosome evolution)?)

Elsewhere:

All *demonstrating, etc.*, should be rephrased with *suggesting, indicating* or similar.

Reviewer #3 (Remarks to the Author):

I have no further comments.

Reviewers' comments:

Reviewer #2 (Remarks to the Author):

Finding the genetic basis for speciation is not a trivial task and somewhat of the Holy Grail in speciation research. It is impossible to get conclusive evidence regarding processes (i.e. speciation) from genomic patterns. In this article, the authors make too strong statements about causality. I strongly disagree that these results provide evidence that this neo-sex chromosome (which is much older than the speciation event) drives speciation. Also, they draw too strong emphasis on the neo-Y: the neo-X and to some extent ancestral-X are also highly differentiated (yes, I know, not as highly as neo-Y, but nevertheless a DM-incompatibility may act on just a few genes or SNPs, and you cannot dismiss all potential interactions involving neo-X or ancestral-X with your data). What you have found however is that strong differentiation of sex chromosomes coincides with reproductive isolation. To me this is an amazing finding that could be followed up in future research (e.g. with functional genomics of sex chromosome-linked genes; knockouts, etc.) to reveal the potential causality between sex chromosome-linked genes and speciation. I am impressed by the data and results in this paper, but unimpressed by the unnecessary overstatements in the title and abstract and elsewhere.

Response: We believe that there is considerable overlap between our interpretation of these data and the interpretations of the reviewer. We certainly are not trying to overstate our findings. We agree that the results highlight several intriguing aspects of neo-sex chromosome evolution and speciation in this important system. We also agree that while sex-linked reproductive isolation is discernible from our extensive crossing and population genomic data, the results remain indirect with respect to the causative roles of the neo-Y, neo-X, and the ancestral X in reproductive isolation.

We believe that we have presented a thoughtful and balanced view of these inferences and uncertainties. We simply would be remiss not to highlight the incredibly compelling patterns on the neo-Y that are consistent with a central role in reproductive isolation (and to not acknowledge the comparably weak evidence relative to the ancestral X). But we certainly do not dismiss the potential role of the X. Indeed, we clearly highlight the fact that interactions between the neo-X and neo-Y are highly plausible routes to the evolution of reproductive isolation. As we thoroughly outlined in our previous revision, we added a more complete discussion of the genetic architecture of reproductive isolation, including a more balanced treatment of the possible contributions of the neo- and ancestral X chromosomes throughout and the contributions of related X-linked dynamics (e.g., dominance theory). See LN 102-105; 275-278; 327-332; 429-438; 452-456; L457-467. As we seem to agree with the reviewer in this respect, we have retained these discussions of the potential role of the X in the final revision.

Title, should read:

Recent sex chromosome evolution *coincides* with speciation in a major forest pest

Response: We had always interpreted use of “coupled” as a statement of a compelling association between two remarkable patterns in our data – the extraordinarily evolution of reproductive isolation and the striking patterns of neo-sex chromosome evolution specialization,

degeneration, and population differentiation. We did not consider this a statement of an outright demonstration as would be necessary with additional genetics and/or direct manipulation. However, given the ambiguity pointed out by the reviewer, we are fine with changing the title. Indeed, we had already offered to the editor to do this in our previous revision. However, we feel that the reviewer's suggested version is only partially representative of the work as it focuses the result entirely on the putative connection to speciation and not more generally on the striking aspects patterns of reproductive isolation and neo-sex chromosome evolution that we do indeed demonstrate.

Therefore, we suggest the following:

“Rapid neo-sex chromosome evolution and incipient speciation in a major forest pest”

Abstract; the final sentence should be rephrased:

Our result *suggests* that neo-sex sex chromosome evolution *may* drive rapid...

Response: We have edited the sentence to include the verb *suggests*.

(By the way, this neo-sex chromosome was formed long before this particular speciation event, but you state in the final sentence ...that early stages of neo-sex chromosome evolution can drive... You do not mean *early* I assume (or perhaps you mean early states of sex chromosome evolution rather than early stages of neo-sex chromosome evolution)?)

Response: We have removed “the early stages of” for simplicity but we also feel that this is just semantics at this point. We have been very thorough about the exact stage of (neo) sex chromosome evolution that we are considering, as outlined in Figure 1 and as discussed throughout the paper. Most previous studies that we have read on the early stages of neo-sex chromosome evolution refer to the initial process of specialization and degeneration as neo-sex chromosome evolution. This term is not reserved only to the establishment phase of a new sex chromosome system. We have simply followed this established practice throughout the manuscript.

Elsewhere:

All *demonstrating, etc.*, should be rephrased with *suggesting, indicating* or similar.

Response: We have edited all remaining uses of the word *demonstrate* in this context. In addition to the abstract, this includes one additional statement in the Discussion: Line 442: “Our results suggest that sex chromosome-autosome fusions can initiate rapid divergence between populations and that this divergence is likely to have fitness consequences in hybrids”.

Reviewer #3 (Remarks to the Author):

I have no further comments.